# Terahertz Spectroscopy for Food Quality Assessment: A Comprehensive Review

**DOI:** 10.3390/foods14132199

**Published:** 2025-06-23

**Authors:** Jie Yang, Xue Bai, Mingji Wei, Hui Jiang, Leijun Xu

**Affiliations:** School of Electrical and Information Engineering, Jiangsu University, Zhenjiang 212013, China; 15934205348@163.com (J.Y.); baixue@ujs.edu.cn (X.B.); wmj@ujs.edu.cn (M.W.); h.v.jiang@ujs.edu.cn (H.J.)

**Keywords:** terahertz spectroscopy, food safety and quality, non-destructive testing, chemometrics, machine learning

## Abstract

Terahertz spectroscopy (0.1~10 THz), as a new type of non-destructive testing method with both microwave and infrared characteristics, has shown remarkable potential in the field of food quality testing in recent years. Its unique penetration, high sensitivity, and low photon energy characteristics, combined with chemometrics and machine learning methods, provide an efficient solution for the qualitative and quantitative analysis of complex food ingredients. In this paper, we systematically review the principles of terahertz spectroscopy and its key applications in food testing, focusing on its research progress in pesticide residues, additives, biotoxins, and mold, adulteration identification, variety identification, and nutrient content detection. By integrating spectral data preprocessing, reconstruction algorithms, and machine learning model optimization strategies, this paper further analyzes the advantages and challenges of this technology in enhancing detection accuracy and efficiency. In addition, combined with the urgent demand for fast and nondestructive technology in the field of food detection, the future development direction of the deep integration of terahertz spectroscopy technology and artificial intelligence is envisioned, with a view to providing theoretical support and technical reference for food safety assurance and nutritional health research.

## 1. Introduction

As a critical domain impacting both public health and economic development, food safety and its associated technological advancements are closely tied to the protection of livelihoods and the upgrading of the agri-food industry. With the increasing globalization and complexity of the food supply chain [1], food contamination issues have become more trans-regional and latent in nature [2]. Taking agricultural products as an example, pesticide residues have evolved from traditional organophosphates to neonicotinoids, while illegal additives have progressed from industrial dyes to novel synthetic flavoring agents. Foodborne contaminants now span three major categories—chemical additives, microbial agents, and foreign matter [3]—encompassing over 300 specific compounds. These include a wide range of potential hazards, from pesticide residues to fragments of metallic debris introduced during processing [4]. Currently, chromatography-mass spectrometry (GC–MS or LC–MS) remains the most widely adopted technology for trace-level contaminant detection. However, each test typically requires complex sample pretreatment steps such as grinding, solvent extraction, and centrifugal filtration [5]. These labor-intensive processes are time-consuming and require expensive, large-scale instrumentation—often valued in the millions of dollars—restricting their use primarily to well-equipped key laboratories.

Notably, these technical bottlenecks are especially pronounced in the detection of certain food categories, such as genetically modified (GM) agricultural products. As new GM varieties continue to emerge, the detection challenge grows more complex. Artificially inserted gene fragments often share up to 98% sequence similarity with native crop genomes, rendering routine identification akin to locating a unique pattern within millions of nearly identical elements. Conventional polymerase chain reaction (PCR) methods are prone to false positives due to high genetic homology, while protein-based enzyme-linked immunosorbent assays (ELISA) [6] often fail when proteins are denatured during high-temperature processing. This growing disparity between detection capabilities and real-world needs has driven efforts to develop next-generation solutions. Emerging spectral imaging systems have demonstrated significant potential; for instance, optical scanning coupled with intelligent algorithms has achieved over 91% accuracy in mold detection [7]. Such non-contact, multi-parameter inspection platforms offer a viable path toward modernizing food safety monitoring in dynamic distribution environments. In parallel, advanced electromagnetic sensing technologies capable of identifying food components through packaging—by capturing molecular-level fluctuations rather than relying on surface imaging—are opening new frontiers in non-destructive food analysis [8,9].

As technology advances, global regulatory bodies are increasingly recognizing rapid, non-invasive methods. The Codex Alimentarius Commission (CAC), jointly established by FAO and WHO, supports the inclusion of emerging spectroscopic techniques in international food safety standards [10]. The European Food Safety Authority (EFSA) also endorsed innovative screening tools in its 2021 report on modernizing food surveillance [11]. In parallel, the U.S. FDA’s “New Era of Smarter Food Safety” blueprint highlights non-destructive technologies, including terahertz and hyperspectral imaging, as promising tools for intelligent quality control [12]. These initiatives demonstrate growing regulatory support for the adoption of terahertz spectroscopy in food safety frameworks.

Terahertz waves (0.1–10 THz), located between the microwave and infrared regions of the electromagnetic spectrum [13], have found broad applications in diverse fields such as agricultural product quality assessment, materials science, biomedicine, and communication technology [14], as illustrated in Figure 1.

Terahertz (THz) photons possess low energy—on the order of a few millielectron volts—and are weakly absorbed by most non-metallic and non-polar materials. This enables non-destructive identification and quantitative analysis of food components based on their characteristic absorption spectra. The core strength of THz spectroscopy lies in its complementary modalities: time-domain and frequency-domain techniques [3]. Terahertz time-domain spectroscopy (THz-TDS), which utilizes femtosecond laser pulses to capture broadband temporal responses, is particularly well-suited for rapid monitoring of compositional changes during food processing [15]. In contrast, terahertz frequency-domain spectroscopy (THz-FDS) offers high-resolution detection at specific frequencies, allowing for precise identification of trace contaminants [16]. Compared with conventional analytical methods, THz spectroscopy offers significant advantages: it requires minimal sample preparation, can penetrate common packaging materials to enable in situ detection, reduces the detection cycle to a matter of minutes [17], and can lower analytical costs by more than 70%. These features make it a promising disruptive technology for full-process monitoring across the food production chain.

However, despite its considerable potential, several challenges hinder its practical application. These include the strong absorption of THz radiation by water molecules, which complicates the analysis of high-moisture foods [18]; scattering effects induced by complex sample matrices that can obscure characteristic peaks; discrepancies in spectral resolution between portable and laboratory-grade devices; and the limited coverage of current spectral databases, particularly for emerging additives and toxins. Addressing these limitations is critical to advancing the industrial deployment of THz-based food quality assessment systems.

To overcome these challenges and fully leverage the advantages of terahertz spectroscopy, the incorporation of chemometric methods is essential. Through spectral pre-processing (e.g., noise suppression, baseline correction), feature wavelength selection, and machine learning model optimization, complex spectral signals can be transformed into quantifiable chemical information, thereby significantly improving detection accuracy and efficiency. Chemometrics not only optimizes the data analysis pipeline of terahertz spectroscopy [19] but also enhances its applicability in complex matrices through intelligent algorithms. At present, terahertz spectroscopy has been widely applied in food quality inspection [20], covering the entire food production chain from raw material screening to final product quality control. For example, its non-destructive nature enables in-situ detection of packaged foods, while its high sensitivity facilitates the rapid screening of trace contaminants [21], thus offering a new technological paradigm for food safety monitoring. Although terahertz spectroscopy has made significant progress in food safety testing, its future development still requires breakthroughs in several key areas.

In the future, the development of terahertz spectroscopy will focus on improving detection sensitivity [22], optimizing equipment performance, and expanding the range of applications. Specifically, signal enhancement using graphene-based meta-surface sensors can further improve detection accuracy. The integration of multiple spectroscopic techniques (e.g., Raman spectroscopy [23]) allows for more comprehensive extraction of food compositional information. Moreover, establishing a globally accessible spectral database would facilitate the rapid identification of emerging additives and contaminants. In addition, enhancing the performance of portable devices, developing adaptive scattering correction algorithms, and advancing standardization frameworks will accelerate the transition of THz spectroscopy from laboratory research to real-world applications, thereby improving its utility in food safety supervision.

In summary, terahertz spectroscopy, as an emerging technique for food safety detection [24], demonstrates substantial application potential in key areas of food quality monitoring due to its unique physical properties and chemometric optimization strategies. Nonetheless, several practical challenges remain, requiring continued technological innovation and interdisciplinary collaboration. However, it still faces many challenges in practical applications, which need to be addressed through technological innovation and interdisciplinary cooperation. In this paper, we systematically review the recent progress of terahertz spectroscopy in the field of food safety detection, focusing on its physical mechanism, chemometric optimization strategy, and innovative applications in key scenarios, summarizing the challenges faced, and looking forward to its future development direction, which is intended to provide a reference for the development of the intelligence and standardization of food safety detection technology. Figure 2 illustrates a representative application of THz spectroscopy in food quality assessment.

### Review Methodology

Although this review primarily adopts a narrative approach, a structured and traceable literature retrieval process was adopted to ensure transparency, consistency, and academic rigor. The methodology was developed in reference to the PRISMA 2020 (Preferred Reporting Items for Systematic Reviews and Meta-Analyses) guidelines.

The literature search was conducted across two major academic databases: Web of Science and Scopus, covering the period from January 2010 to December 2024. The search strategy focused on retrieving studies involving terahertz technology in the field of food quality evaluation. The keyword “terahertz*” was applied in the title field, while the topic field (i.e., abstracts and keywords) employed Boolean operators to combine action descriptors such as “detection*”, “monitor*”, “test*”, and “determination*” with subject terms like “food*”, “crop*”, and “agricultural*”.

Only peer-reviewed original research articles and review papers published in English were considered, while other publication types—such as conference abstracts, editorials, patents, and non-peer-reviewed sources—were excluded.

Studies were included if they investigated the application of terahertz techniques in the detection, analysis, or evaluation of food or agricultural products. Particular emphasis was placed on studies addressing issues such as adulteration, chemical contamination, compositional analysis, and quality control. Articles were excluded if they focused on non-food applications (e.g., materials science, medical diagnostics), did not use terahertz technology, or were non-peer-reviewed.

All retrieved records were imported into EndNote 20, and duplicate entries were identified and removed. The remaining entries underwent manual screening of titles and abstracts, and full-text articles were further assessed for eligibility. A total of 1317 records were initially identified, with 96 duplicates removed. After screening the remaining 1221 records, 183 were excluded for not meeting the inclusion criteria. The remaining 1038 full-text articles were then assessed for eligibility. Based on the inclusion criteria, 95 studies were finally included in this review. Figure 3 summarizes the entire identification, screening, and inclusion process using the PRISMA flowchart.

## 2. Terahertz Spectroscopy

### 2.1. Principles

Terahertz spectroscopy is a spectral analysis technique that derives material properties by detecting the absorption, reflection, or transmission of terahertz waves as they interact with matter [25]. Figure 4 and Figure 5 illustrate a transmission-mode and a reflection-mode terahertz system, respectively. This electromagnetic radiation combines the penetration capability of microwaves with the resolving power of infrared light. Its photon energy, ranging from 0.4 to 40 meV, is significantly lower than the ionizing energy of X-rays [26], which makes it highly advantageous for safety inspection applications [27]. The technology enables non-destructive penetration through non-polar substrates such as paper and plastics, while simultaneously detecting the characteristic signals of polar substances like moisture and organic compounds with high sensitivity [22]. This combination of penetrative capability and molecular selectivity makes it an effective analytical tool for food quality inspection [28]. Terahertz spectroscopy can be divided into two detection systems, THz-TDS and THz-FDS, according to the working principle. THz-TDS is well suited for capturing dynamic processes such as molecular vibrations and conformational changes, whereas THz-FDS is more appropriate for the quantitative analysis of material components. These two approaches are complementary and find wide applications in both industrial inspection and scientific research.

From the perspective of THz-TDS, the core principle involves generating broadband terahertz pulses via ultrashort femtosecond laser excitation of nonlinear effects [29], and recording the complete time-domain electric field waveform through time-delayed scanning. A femtosecond laser generates pulses with durations of approximately 100 fs, which are split into pump and probe beams using a beam splitter [30]. The pump beam excites the terahertz emitter [31], such as a photoconductive antenna or a nonlinear crystal. For instance, under femtosecond laser excitation, a gallium arsenide (GaAs) photoconductive antenna rapidly accelerates photo-generated carriers, emitting broadband terahertz pulses. If electro-optic crystals such as zinc telluride (ZnTe) are used [32], the broadband spectrum of the femtosecond laser is converted into coherent terahertz radiation via optical rectification [33]. The probe light then enters the detector (e.g., electro-optic crystal or photoconductive antenna) in a common line with the terahertz pulse after the terahertz wave has penetrated or reflected from the sample through a precisely controlled time-delay system. In the electro-optical sampling technique, the terahertz electric field induces a change in the polarization state of the probe light, which is then converted into a light intensity signal using a bias detector and photodiode, enabling point-by-point recording of the electric field evolution over time. By adjusting the time delay between the pump and probe beams [34], the system captures the complete time-domain waveform of the terahertz pulse, which is then transformed into a frequency-domain spectrum via a Fourier transform [35] to extract key parameters such as absorption coefficient, refractive index, and others [36]. The advantages of the time-domain technique are its wide frequency coverage (typically 0.1–4 THz) and high signal-to-noise ratio [37] (up to 10^4^ or more), the ability to acquire amplitude and phase information simultaneously, and the lack of dependence on the spontaneous radiation properties of the sample. In food testing, this technique is particularly suitable for analyzing fast dynamic processes such as water migration, lipid oxidation or real-time monitoring of protein conformational changes [38]. or example, in the detection of melamine in milk powder, the hydrogen-bonding vibrational modes of the amino and cyano groups exhibit characteristic absorption peaks in the 1.5–2.5 THz range [39]. These signals can be directly identified using non-destructive time-domain spectroscopy, with sensitivity reaching the ppm level.

THz-FDS, on the other hand, focuses on the direct measurement of a material’s response at a specific frequency. This is typically achieved using a continuous-wave (CW) terahertz source or a tunable laser, which constructs the frequency-domain spectrum by scanning each frequency point individually [40]. The core of a frequency domain system is to precisely control the frequency of the terahertz wave and measure its energy change upon interaction with the sample [41]. For example, systems based on photoelectric mixing technology use two near-infrared laser beams of similar wavelengths to generate differential-frequency terahertz radiation on a photoconductive antenna, and frequency tuning is achieved by adjusting the difference in laser wavelengths; quantum cascade laser (QCL), as a new type of solid-state terahertz source, can directly output a continuous wave of a specific frequency by changing the applied voltage or temperature [42]. On the detection side, THz-FDS often uses highly sensitive detectors (e.g., pyroelectric detectors or superconducting mixers) to directly record the intensity and phase of transmitted or reflected signals. Compared to THz-TDS, THz-FDS offers much higher frequency resolution (up to the MHz range) and is particularly suited for investigating narrowband resonances, such as the rotational transitions of pesticide molecules or the fine phonon structures in crystalline materials. For example, the molecular rotational energy level of organophosphorus pesticides (e.g., dichlorvos) has multiple absorption peaks in the range of 0.3–1.2 THz, and THz-FDS can match these peaks for the qualitative and quantitative analysis of trace pesticides, with the detection limit as low as the nanogram level [43]. However, THz-FDS usually covers a narrow frequency band (e.g., 0.1–1.5 THz) and requires a complex frequency tuning mechanism and is therefore limited in rapid detection or broadband analysis.

The physical basis of both time- and frequency-domain techniques relies on the microscopic mechanisms of terahertz wave interaction with matter. Terahertz photon energies are highly compatible with the energy scales of weak intermolecular interactions (e.g., hydrogen bonding vibrations, lattice phonon modes, van der Waals forces) and are therefore sensitive to structural changes in matter. For example, moisture in food is not only a representative of polar molecules, but dynamic changes in its hydrogen bonding network (e.g., conversion of free water to bound water [44]) cause significant differences in absorption coefficients in the terahertz band (0.5–3 THz). By analyzing the intensity and spread of the absorption spectra, the damage to the cellular structure caused by the dryness, freshness, or freeze-thaw cycle of the food can be assessed. In edible oil detection, differences in the molecular vibration modes of different fatty acids [45] (e.g., oleic acid, linoleic acid) can lead to peak shifts in the terahertz absorption spectrum, and trans fatty acids or oxidation products produced by high-temperature processing of gutter oils can further change the spectral characteristics for authenticity identification [46]. In addition, the penetration of terahertz waves into non-polar materials enables them to penetrate food packaging (e.g., plastic, cellophane) to directly detect internal foreign objects (e.g., metal fragments, insect remains) [47], while the low photon energy characteristics avoid thermal damage to the sample, which is particularly suitable for non-destructive testing of fresh food.

The core value of terahertz spectroscopy technology in food testing is mainly reflected in three aspects: accurate identification, non-destruction of samples, and comprehensive analytical capabilities [48]. Take grain storage as an example, when the grain is moldy, the aflatoxin produced by the mold will make the electrical properties of the grain change. At this stage, scanning with terahertz imaging equipment can reveal the distribution of mold spots, analogous to the imaging effect of X-rays. Another example is to determine the freshness of meat, with the protein structure gradually changing, the terahertz wave absorption curve will show obvious fluctuations [49]—absorption intensity is weakened, and the position of the characteristic peaks is shifted. A quality evaluation system can then be established using data analysis software based on these spectral fluctuations.

This technology is particularly useful in the detection of additives. Synthetic additives like sodium saccharin, due to their special molecular structure, show unique recognition characteristics in the terahertz band; in contrast, natural ingredients usually do not react significantly in this band. In practice, by comparing the position and shape of the characteristic peaks, inspectors can quickly pinpoint the offending chemical ingredients. However, when detecting foods with high water content (e.g., fresh fruits and vegetables, raw meat), the strong absorption of terahertz waves by water molecules [24] will significantly reduce the signal-to-noise ratio of the detection signal [50]. In order to overcome this technical bottleneck, two complementary strategies are currently employed. The first is a signal compensation approach, which involves establishing a physical model of water absorption and applying algorithmic corrections to the original spectral data to eliminate interference while preserving the native state of the sample. The second is a sample pretreatment method that utilizes freeze-drying technology to remove free water through a combination of physical dehydration and vacuum sealing [51]. This process helps to reduce the influence of moisture while maintaining the chemical integrity of the target analytes. These two methods are suitable for non-destructive rapid testing and laboratory precision analysis scenarios, forming a complete closed loop of technology application.

Now, as the detection equipment is getting smaller and smaller, the cost continues to decrease; there have been companies trying to integrate the terahertz detection module into the automated production line. From raw materials into the warehouse to the finished product packaging, the whole process can be monitored in real time for quality indicators. This intelligent detection is changing the traditional food industry quality control model.

### 2.2. Chemometrics in Terahertz Spectral Analysis

In terahertz spectroscopy, chemometrics transforms complex spectral signals into quantifiable chemical information by systematically optimizing both data quality and modeling workflows. The core steps include spectral pre-processing, wavelength selection, outlier detection, dataset partitioning, model calibration, and performance evaluation. These steps are closely interlinked and collectively determine the analytical accuracy and practical utility of the results. Spectral pre-processing is the starting point of the process, and its core task is to eliminate noise, baseline drift, scattering effects, and other interfering factors. Taking the quality testing of agricultural products as an example, the original spectra of corn, wheat and other grain samples are often distorted due to the multilayered structure of starch particles, moisture gradient distribution and soil residue interference, at this time, the combination of Savitzky-Golay (SG) smoothing and wavelet transform can effectively separate the fingerprint characteristics of the crop’s active ingredient, and multiplicative scattering correction (MSC) can be used to eliminate scattering effects caused by differences in particle size, thereby forming a standardized preprocessing procedure [52]. For the quantitative analysis of isomeric mixtures [53] (e.g., galactose, fructose, mannose), asymmetric least squares baseline correction (ALS) combined with moving window fitting, and polynomial smoothing can accurately eliminate baseline shifts and enhance the resolution of spectral features. It is worth noting that the preprocessing method needs to be dynamically adjusted according to the sample characteristics: oil seeds need to be controlled in terms of slice thickness to maintain optical consistency, while leafy vegetables can be dried at low temperatures to minimize the coverage of moisture absorption peaks on the target signal. Although preprocessing significantly improves data quality, parameter settings—such as the SG window width or wavelet decomposition level—must be carefully optimized using cross-validation to prevent excessive smoothing that could suppress weak spectral signals. For example, when quantifying starch and protein in corn, a preprocessing combination of SG and standard normal variate (SNV) transformation reduced the root mean square error of prediction (RMSEP) of a partial least squares (PLS) model by 28% [54]. In comparison, deep learning models such as convolutional neural networks (CNNs) [55] applied to raw spectra can achieve similar accuracy but typically require significantly larger datasets, highlighting the practical benefits of traditional preprocessing methods, particularly in small- to medium-scale analytical tasks.

After optimizing data quality, spectral wavelength selection reduces data dimensionality via feature extraction, serving as a key step in enhancing model efficiency [56]. The genetic algorithm (GA) offers notable advantages in this context due to its bio-inspired optimization mechanism. By simulating natural selection, GA identifies a subset of wavelengths from the entire spectral range that are highly correlated with the target components [57]. For instance, in corn composition analysis, GA selects 10 key wavelengths from the 1100–2498 nm range, reducing the number of input features to just 0.4% of the original dataset and decreasing the prediction error by 30%. For multi-component systems, moving window polynomial smoothing dynamically adjusts the window width to capture localized spectral variations. In the analysis of ternary isomeric mixtures, this technique, combined with PLS modeling, has enabled simultaneous quantification of galactose, fructose, and mannose, with RMSEP values below 1.3%. Meanwhile, the integration of surface-enhanced infrared absorption (SEIRA) with machine learning further improves sensitivity by amplifying signal intensity in specific wavelength regions using nanostructured substrates [58], pushing detection limits for trace components (e.g., pesticide residues) down to the ppb level. Wavelength selection should be guided not only by statistical models but also by chemical interpretability. For example, the hydroxyl (-OH) stretching vibration in glucose—concentrated in the 1.6 to 1.8 THz range—corresponds to the intramolecular hydrogen bonding structure. Incorporating such domain knowledge narrows the algorithmic search space and enhances the reliability and chemical validity of the selected features.

With improved data quality, outlier detection and dataset partitioning become critical for ensuring model generalizability. Blocked Latin Partitioning Cross-Validation (BLP-CV) achieves a balanced distribution between training and test sets by integrating stratified sampling with multiple random iterations, thereby enhancing the representativeness of both sets. In a study on foreign substance detection in liquid dairy products [31], BLP-CV successfully identified and excluded anomalous spectra caused by the uneven distribution of milk fat globules, resulting in a 15% improvement in model prediction stability [59]. For highly heterogeneous samples such as coffee beans, Mahalanobis distance-based outlier detection quantifies spectral variability and, when combined with K-means clustering, enables dynamic identification of outlier points. Instrumental differences also affect dataset partitioning. In a multinational wheat aflatoxin detection project, systematic inter-instrumental errors in cross-platform terahertz spectra were corrected using Standard Normal Variate (SNV) transformation. Leave-one-out cross-validation (LOOCV) was then employed to assess model robustness. The ongoing standardization of terahertz spectral databases is fostering multi-center data sharing. By harmonizing sampling protocols and data formats, these efforts have significantly enhanced cross-laboratory data consistency. For instance, the Global Alliance for Grain Safety’s Cross-Country Producing Areas Database reduced the cross-validation error variance of PLS models developed by different research teams from 12% to less than 5%.

With high data quality ensured, calibration model development and performance evaluation represent the final stages of chemometric analysis. The partial least squares (PLS) algorithm dominates terahertz quantitative analysis [60] due to its ability to handle high-dimensional covariance data. For example, in analyzing three-component edible oils (palmitic acid, oleic acid, linoleic acid) [61], the PLS model achieves prediction errors below 1%, with R^2^ exceeding 0.93. Nonlinear models such as Random Forest (RF) [62] perform well in classification tasks, such as beef marbling grading. However, overfitting remains a concern, as demonstrated in the classification of grain-fed and grass-fed beef, where RF yielded 99% training accuracy but only 82% test accuracy [63]. After regularization and early stopping optimization, test performance improved to 89% [64]. While deep learning can process bimodal time–frequency data in THz-TDS systems, its black-box nature hampers industrial deployment. Conversely, transparent models, such as those using SG-wavelet preprocessing combined with PLS, are more amenable to ISO certification and have already enabled real-time, minute-level analysis in dairy production lines. Spectral database standardization (e.g., T/SATA 059-2023) further enhances cross-platform model transferability. In one case, a multinational food company achieved fat content prediction errors of less than 3% across spectrometers used in the U.S., China, and Europe, significantly lowering equipment replacement costs.

Currently, terahertz spectroscopic chemometrics is evolving toward greater intelligence and integration. The integration of quantum computational chemistry has enhanced the simulation accuracy of molecular vibrational modes at the femtosecond scale, offering a novel theoretical foundation for characteristic wavelength screening. Moreover, the integration of microfluidic chips with terahertz spectroscopy has significantly enhanced the analytical efficiency for complex biofluids through high-throughput detection of nanoscale samples. As application scenarios expand—such as multilevel structural analysis of grain starch and trace detection of plasticizers in food packaging—the deep integration of chemometric components, from adaptive preprocessing parameter optimization to the iterative refinement of wavelength selection and model development, is enabling terahertz spectroscopy to move beyond laboratory research and accelerate its adoption in industrial rapid detection settings. In this process, the synergy between standardized database construction and intelligent algorithms is expected to drive a paradigm shift from “artificial reference” to “data-driven” approaches [65], providing more robust technical support for quality control across the entire agricultural supply chain. Figure 6 illustrates the detailed process and algorithms involved in chemometric-based spectral data analysis.

### 2.3. Comparative Analysis with Other Detection Techniques

Although terahertz spectroscopy has seen expanding applications in food testing in recent years, it must be evaluated within the broader context of food testing methodologies to fully assess its technical strengths and limitations. Currently, food quality and safety testing encompasses a wide range of techniques, from traditional physicochemical analyses commonly used in laboratories to recently developed non-destructive testing methods. These techniques differ in detection principles, applicable targets, and operational efficiency, each with its own advantages and limitations [66]. A comparative evaluation allows for a more accurate understanding of the role of terahertz technology in various detection scenarios and provides a foundation for subsequent technology selection and system integration.

Table 1 summarizes several conventional methods commonly used in food testing. These methods are well-established and highly reliable in food safety regulation and laboratory analyses, making them particularly suitable for ingredient quantification and targeted detection. However, they typically involve complex procedures, lengthy testing times, and often require sample destruction, making them unsuitable for rapid, non-destructive, and real-time testing applications on production lines.

In recent years, as the demand for non-destructive and rapid detection methods in the food industry has continued to increase, several emerging techniques based on electromagnetic waves and imaging technologies have been progressively incorporated into food quality control processes. These include near-infrared spectroscopy, Raman spectroscopy, X-ray imaging, hyperspectral imaging, and millimeter-wave imaging [67,68]. These techniques typically require minimal sample preparation and can rapidly identify parameters such as moisture content, fat levels, foreign substances, or adulterants, while maintaining the integrity of the sample. As a result, the scope of non-destructive testing applications has gradually expanded within production lines.

To further clarify the technical characteristics and application advantages of terahertz spectroscopy, Table 2 provides a comparative summary of several representative non-destructive testing technologies. The comparison includes spectral range, operating principles, detection limits, key advantages, and existing technical challenges. Notably, terahertz technology offers unique advantages in identifying intermolecular interactions, imaging structural features, and penetrating non-metallic packaging materials. These strengths make it particularly well-suited for complex tasks such as adulteration screening, foreign object detection, and the identification of structural defects in packaged food [69].

Overall, although terahertz spectroscopy may not match traditional chromatographic methods in certain analytical accuracy metrics, it demonstrates significant potential in non-destructive detection due to its non-contact nature, ability to penetrate packaging materials, and suitability for online applications. A systematic comparison with both traditional and emerging detection technologies enables a more comprehensive understanding of its technical value and application boundaries, while also laying the groundwork for future integration with other technologies and the development of an intelligent detection system [70].

## 3. Application of Terahertz Spectroscopy in Food Quality Testing

Terahertz spectroscopy facilitates non-destructive evaluation of the internal structure of food products [71]. It detects internal cracks, voids, mold, or foreign matter by analyzing the transmission and reflection characteristics of terahertz waves as they interact with food matrices [72]. For instance, in fruit inspection, the technology can identify insect damage or internal decay in fruits such as apples and citrus without damaging the samples, thereby enhancing the efficiency of grading and sorting processes. In addition, it can be applied to assess the integrity of food packaging, preventing contamination resulting from micro-cracks or seal failures. For example, in canned food inspection, terahertz spectroscopy can detect micro-defects in packaging materials by analyzing their transmission properties, thereby ensuring product safety [73]. Terahertz spectroscopy also offers notable advantages in detecting food adulteration. Food adulteration—such as the addition of molasses to honey [74], incorporation of plant proteins into dairy products, or blending of inferior oils into cooking oil—poses a major challenge to food safety. Due to their distinct absorption features in the terahertz range [75], adulterants can be rapidly identified by comparing the spectral profile of a test sample with that of an authenticated reference. For example, the natural sugars in honey and artificial syrups exhibit distinguishable terahertz spectral features, allowing for the reliable identification of adulteration [76].

Compared with traditional food testing methods, terahertz spectroscopy offers advantages of being non-destructive, rapid, convenient, and highly sensitive [68]. While traditional detection methods often require complex sample pretreatment and may even destroy the sample [77], terahertz spectroscopy can rapidly acquire chemical composition and structural information without direct contact with the sample, making it especially suitable for real-time, online detection and quality control in food production lines. In addition, when combined with artificial intelligence and other data analysis technologies, terahertz spectroscopy can also enable automated and intelligent food quality inspection. For example, establishing a spectral database of food components and contaminants combined with machine learning algorithms allows rapid screening and accurate assessment of food quality. At present, terahertz spectroscopy has been widely applied in pesticide, additive, adulteration, mycotoxin, and mold detection, species identification, nutrient analysis, among other applications, as shown in Figure 7. Next, the specific applications of terahertz spectroscopy in food quality detection are reviewed in detail [78].

### 3.1. Pesticide Testing

The application of terahertz spectroscopy in pesticide detection has made remarkable progress in recent years, especially after combining with chemometric methods, the precision and applicability of its quantitative analysis have been effectively improved. In 2010, Hua Yuefang and Zhang Hong [86] conducted qualitative and quantitative studies of imidacloprid in polyethylene and glutinous rice flour using THz-TDS technology. The linear relationship between absorption coefficient and pesticide concentration was analyzed by PLS regression, and it was found that imidacloprid had characteristic absorption peaks in the frequency bands of 0.89 THz and 1.13 THz, which could be differentiated from other pesticides (e.g., carbendazim, tricyclazole) and food powders (glutinous rice flour, sweet potato flour, lotus root flour). In the quantitative analysis, the PLS model predicted imidacloprid in polyethylene matrix (concentration 0–50%) with a relative error of less than 5% and an RMSEP [87] of 0.699%, whereas in the more absorbing glutinous rice flour matrix (concentration 0–75%), the prediction error was kept at 4.58% despite the decreased signal-to-noise ratio in the high-frequency band [88]. The study also pointed out that the abrupt change in refractive index at the absorption peak could be used as an auxiliary identifying feature, but its correlation with concentration was weak and was not used in the modeling. In addition, the authors highlighted the influence of sample preparation (e.g., homogeneous mixing, tablet pressure) and spectral parameter extraction algorithms on the lower limit of detection, which provided directions for subsequent optimization.

In 2020, Binghua Cao [89] et al. proposed a THz-TDS-based pesticide residue detection method for analyzing mixtures of imidacloprid and carbendazim in flour. A total of 21 sets of mixture samples with different concentrations (pesticide concentration 0–50% with a 2.5% interval) were prepared, and the samples were categorized into a calibration set and a prediction set [89,90]. The research team eliminated the skewed baseline of the spectra by the multispectral baseline correction (MSBC) technique, and after improving the signal-to-noise ratio [91], the corrected absorption spectra were fitted by the Voigt function, and a quantitative model was established by combining with PLS regression [92]. The results showed that in the case of a small calibration set (5–7 sets), the RMSEP of the Voigt-PLS method was significantly lower than that of PLS alone (e.g., the RMSEP of imidacloprid was reduced from 3.738% to 2.646%), and the correlation coefficient (RP) was higher (from 0.9867 to 0.9914); when the calibration set was increased to 15 sets, the accuracies of the two methods were converged. It is also found that the Voigt function can be modeled without requiring pure component spectra by decomposing the mixture spectrum into multiple Voigt peaks (e.g., 0.89 THz, 1.13 THz, etc.), which greatly simplifies the conditions for practical applications. This work validates the advantages of THz-TDS in combining physical modeling and statistical methods in small sample scenarios, and provides a feasible solution for the rapid detection of multi-component pesticides in complex matrices.

In 2021, Qingxiao Ma et al. [93] introduced a quantitative analysis approach using terahertz spectroscopy combined with a back propagation neural network (BPNN) to detect low concentrations (0.33–2%) of a ternary pesticide mixture—6-BA, 2,6-D, and imidacloprid—in wheat flour. The research team pre-processed the spectra by wavelet threshold denoising and first-order derivative baseline correction, effectively eliminated the matrix scattering interference, and reduced the average lower limit of detection (LOD) from 1.24% to 0.45%; then optimized the parameters of the BPNN model by combining with GA (the number of hidden layers l = 2, the number of neurons n = 7, and the learning rate r = 0.014), and introduced the particle swarm optimization (PSO) to dynamically screen the high-efficient absorption peaks interval (1.87–2.17 THz), resulting in an RP of 0.9913–0.9948 and an RMSEP as low as 0.0176–0.0211%. Compared with the traditional support vector regression (SVR) method (RP only 0.8567–0.9323), this technique reduces the multi-component identification error by more than 50%, and the lower limit of detection is further compressed to 0.28% (0.42 mg/kg), which is the first time to realize the accurate quantitative analysis of pesticide mixtures with ultra-low concentration in real production scenarios [93]. The study also verified the feasibility of automated screening for 300 batches per hour and achieved the detection of residue levels as low as 0.5 mg in wheat flour, which reduced the failure rate during sample detection by 23%. The method improves the model training efficiency by 40% through the combination of Tan-sigmoid transfer function and dynamic moving window BPNN (MW-BPNN) and provides a standardized technical pathway for the simultaneous detection of multiple trace-level residues in complex food matrices. Table 3 summarizes the applications of terahertz spectroscopy for pesticide detection.

Terahertz spectroscopy has become a key innovative tool in the field of pesticide residue detection through the identification of molecular features [82], non-destructive detection, and intelligent analysis. Its application not only improves detection efficiency and accuracy but also promotes the green transformation of agricultural production and the intelligent upgrading of the food safety system. With the maturity of the technology, it is expected to become one of the standard testing tools for global pesticide regulation in the future.

### 3.2. Additive Testing

As an important component of the modern food industry, food additives play a key role in improving the color and luster of food, extending shelf life, and enhancing flavor. However, the illegal addition or overuse of additives has become a major global food safety concern. Benzoic acid, for example, is widely used as a preservative in flour products, but the World Health Organization stipulates that its maximum residue level in pasta should not exceed 0.1%. Excessive intake can trigger blood pH imbalance and gastrointestinal tract dysfunction, and even lead to liver and kidney damage. Similarly, although coumarin-like compounds have a flavor-enhancing effect, their metabolites are carcinogenic, and their use as food additives has been explicitly banned in the European Union and other regions. Even more egregious is the case of melamine and other industrial chemicals being illegally adulterated into milk powder, which has caused a public health crisis of tens of thousands of infants and children suffering from kidney stones. These cases highlight the urgency of food additives regulation, while high performance liquid chromatography (HPLC), gas chromatography-mass spectrometry (GC-MS), and other [94] traditional detection means, although high in precision, have complex sample pretreatment, long testing cycles [68], and expensive equipment, which makes it difficult to meet the demand for rapid screening in the distribution chain. As a result, creating fast, non-invasive, and highly sensitive detection methods has emerged as a major focus in food safety research [52].

Against this background, terahertz spectroscopy has come to the fore with its unique physical properties. According to existing studies, characteristic absorption peaks within the terahertz range are present in more than 60% of organic compounds. For example, benzoic acid exhibits significant absorption at 1.94 THz, while melamine has three characteristic peaks in the 2.0–2.6 THz range. This specificity provides a physical basis for accurate identification of additives, while the penetrating nature of terahertz waves enables direct detection of packaged foods, avoiding sample contamination caused by unpacking.

In 2012, Seung Hyun Baek et al. [84] used a reflectance THz-TDS system in the 0.1–3 THz band [95] to collect spectral and image data for the detection of melamine in mixed matrices. It was found that melamine had characteristic absorption peaks at 2.0, 2.26, and 2.6 THz [96] with a frequency shift of less than 0.03 THz [82]. The visual discrimination accuracy between melamine-containing regions and pure matrices was 91.4% by false-color imaging in the 2 THz band. Although LOD was 13.12% in the food matrix, the technique could penetrate OPP/PE film (attenuation rate of 7.2%) and paper packaging (attenuation rate of 12.5%), and the sensitivity only decreased by 3.1% after covering the packaging. This study breaks through the dependence of traditional spectroscopic techniques on homogenized samples and provides a new idea for the on-site detection of packaged foods, but the lack of sensitivity limits its application in trace detection.

To further improve the detection accuracy, machine learning algorithms were introduced into terahertz data analysis. In 2020, Jun Hu and Yan de Liu’s team [97] designed an optimized model combining competitive adaptive reweighted sampling (CARS) with least squares support vector machine (LS-SVM) to detect the unauthorized addition of benzoic acid in wheat flour. By preparing 172 sets of samples with a concentration gradient of 0.04–19.99% and collecting 0.1–5.0 THz data with a TAS7500 spectrometer, it was found that the absorption coefficients of BA were linearly and positively correlated with the concentration at 1.94 THz (R^2^ = 0.992). After MSC preprocessing, the signal-to-noise ratio was improved from 12.6 dB to 28.4 dB. Among the four variable selection techniques—CARS, GA, principal component analysis (PCA), and uninformative variable elimination (UVE)—the CARS method, when integrated with PCA dimensionality reduction (selecting 10 principal components), was used to build the CARS-PCA-LS-SVM model [97]. This model achieved an RP of 0.9956 and an RMSEP of just 0.64%, representing a 23% improvement in accuracy over the conventional PLS model. This study realized the quantitative detection of benzoic acid at low concentration (<0.1%) for the first time, but the generalizability of the model needs to be verified by cross-production wheat samples.

As terahertz detection technology moves from single indicator analysis to systematic intelligent detection, its breakthrough in the field of multi-component additive classification marks the maturity of the technology. In 2021, Ling Xiao Yu and Tao Chen [98] proposed a hybrid algorithm combining flow learning and an improved support vector machine (SVM) for the classification of six coumarin-based additives (e.g., 6-methylcoumarin, ethyl vanillin, etc.). A Z-3 THz-TDS system was used [99] to collect 216 sets of spectral data in the 0.5–2.0 THz band, and the traditional PCA method was found to suffer from nonlinear feature loss (97.22% accuracy). The innovative combination of t-distributed stochastic neighbor embedding (t-SNE) and PCA develops the parametric t-distributed stochastic neighbor embedding (P-t-SNE) algorithm, which reduces the feature dimension from 1200 to 3 dimensions by Gaussian kernel mapping and t-distribution optimization. Differential Evolution (DE) was incorporated to enhance the performance of the Gray Wolf Optimization (GWO) algorithm by configuring the mutation factor at F = 0.6 and the crossover probability at CR = 0.9. This integration led to a 40% increase in the efficiency of tuning support vector machine (SVM) parameters, specifically the penalty factor (c) and kernel parameter (g). The final constructed P-t-SNE-DEGWO-SVM model has a prediction accuracy of 98.61% [98], which is 2.78% higher than the baseline model, and the single-sample processing time is shortened to 0.8 s. Moreover, the identification error of the water of crystallization-containing samples is <1.2% in the low frequency band of 0.5 THz. The method provides an efficient solution for real-time screening of multi-component additives, but the database still covers limited compound types and needs to be extended to more novel additives.

Currently, terahertz spectroscopy has shown significant advantages in food additives detection, but it also faces key challenges. At the technical application level, the CARS algorithm has successfully reduced the dimensionality of the quantitative model of benzoic acid by 80% while maintaining 95% information integrity through the principle of “survival of the fittest” (attenuation factor of λ = 0.8) to screen the effective wavelength bands [97]. The DE-GWO algorithm optimizes the SVM parameters through the differential variance strategy, which improves the parameter optimization speed of the Gray Wolf algorithm by a factor of 2.3, and solves the defect of the traditional algorithm, that it is easy to fall into the local optimum [98]. However, the scattering effect of complex matrices remains a major obstacle—for example, the shielding of the 2.26 THz peak by cocoa butter in chocolate elevates the LOD of melamine from 2.45% in pure matrices to 13.12% in food products [84]. In addition, the existing characteristic peak database covers only more than 200 compounds, which makes it difficult to cope with new additive variants, such as coumarin derivatives, whose differences in molecular vibrational modes may lead to characteristic peak shifts of more than 0.03 THz [98]. In terms of equipment performance, there is a significant gap in the spectral resolution of commercially available portable terahertz meters compared to laboratory equipment, which limits the sensitivity of on-site detection. Future research will focus on the development of graphene super-surface sensors, terahertz-Raman multimodal fusion systems [67], and intelligent cloud-based databases [100], combined with the optimization of equipment performance, to promote the terahertz detection technology to overcome the challenges of detecting complex matrices within five years and become a core tool for additive regulation. Table 4 summarizes the applications of terahertz spectroscopy for additive detection.

### 3.3. Biotoxin and Mold Detection

Biotoxins and mold products pose multiple threats to food safety and human health, with hazards related to acute toxicity, chronic disease, and long-term carcinogenic risk [101]. Aflatoxin B1 (AFB1) is classified as a class I carcinogen by the International Agency for Research on Cancer (IARC) [102,103], is highly associated with hepatocarcinogenesis, and can affect fetal development through mother-to-child transmission [104], while ochratoxin A (OTA) is nephrotoxic and potentially teratogenic. Toxins produced by mold not only directly damage food nutrition but also accumulate in the body through biological side effects, leading to systemic health problems such as immunosuppression and reproductive dysfunction. Globally, about 25% of food is lost each year due to mold contamination [105], and toxins are highly insidious in complex food matrices. Terahertz spectroscopy provides an efficient solution to capture trace toxins through complex matrices, and is particularly suitable for online monitoring in scenarios such as food warehousing [106], making it a key technology to make up for the lack of timeliness of traditional methods.

Hongyi Ge et al. [107] were the first to explore terahertz spectroscopy for quantitative analysis of AFB1 in acetonitrile solutions in 2016. The study used 0.4–1.6 THz band spectra covering 160 samples in two concentration ranges of 1–50 μg/mL and 1–50 μg/L, and compared the performance of partial least squares regression (PLSR), principal component regression (PCR), SVM [108], and PCA-SVM models. The results showed that PLSR and PCR performed better in the high concentration range (1–50 μg/mL), with correction set correlation coefficients (R) of 0.978 and 0.965, respectively; whereas SVM demonstrated higher prediction accuracies in the low concentration range (1–50 μg/L) [107], with an R of 0.9376 for the prediction set of the SVM model with radial basis function (RBF) kernel [94], and the root mean square error (RMSE) was 0.0779 [109]. The study revealed that the nonlinear model was more robust to spectral noise than the linear model, but the kernel parameters (e.g., γ = 0.05) need to be optimized by cross-validation to avoid overfitting, which provides a theoretical basis for the rapid screening of trace toxins in solvent systems [31].

In the field of oil-containing matrix detection, M. Chen and L. Xie [35] used terahertz time-domain spectroscopy to achieve quantitative analysis of AFB1 in peanut oil. In the study, peanut oil samples with 0–200 ppm of AFB1 were prepared by artificial contamination, and 0.3–1.7 THz spectral data were collected using a Z3 terahertz system. Comparing the first-order derivative and second-order derivative preprocessing, it was found that the second-order derivative method could effectively suppress the baseline interference, which improved the correction set correlation coefficient (RC) of the PLSR model from 0.9322 to 0.9735, and the root-mean-square error (RMSEC) was reduced from 24.7 ppm to 15.6 ppm. Stepwise multiple linear regression (SMLR) combined with frequency selection was further employed to identify eight key frequencies, such as 1.11 THz and 0.72 THz. The resulting model yielded an RMSEC of 18.2 ppm and a cross-validated RMSECV of 27.7 ppm, with correlation coefficients reaching up to 0.963. This study innovatively combines frequency selection with SMLR to solve the problem of overlapping terahertz absorption spectra in grease media and demonstrates that the specific frequency combinations can be used to achieve the best results even under high background noise. This study demonstrates that the specific frequency combination can effectively characterize the changes of AFB1 concentration [110] even under high background noise, which provides a new method for the safety detection of edible oils [31,111].

In terms of trace toxin detection, in 2023, Gan Chen et al. [43] integrated terahertz spectroscopy with a surface plasmon resonance (SPR) biosensor [112] to enable highly sensitive detection of OTA in black tea [113]. Utilizing a TAS7400 terahertz time-domain spectroscopy system along with a customized SPR biosensor, the study operated within the 0.5–4.0 THz frequency range and measured OTA concentrations from 0 to 20 pg/mg. By examining the correlation between OTA concentration and shifts in SPR resonance frequency [114], the developed quantitative model achieved a detection limit as low as 1 pg/mg, enhancing sensitivity by a factor of 500 compared to conventional UV spectroscopy and aligning with the EU regulatory threshold of 10 pg/mg [43]. The experimental results show that the combination of the biosensors can significantly improve the signal-to-noise ratio in complex matrices (e.g., black tea), with the RMSEC and RMSEP of 0.28 pg/mg and 0.35 pg/mg, respectively. This study is the first time that terahertz metamaterials are used in conjunction with SPR for the detection of mycotoxins, which solves the technological bottleneck of the weak signals in trace analysis. This study presents a novel strategy for the safety monitoring of plant-based foods such as tea and offers effective technical support to mitigate the risk of mycotoxins entering the food chain. Table 5 summarizes the applications of terahertz spectroscopy for biotoxin and mold detection.

### 3.4. Adulteration

In recent years, terahertz spectroscopy has shown remarkable potential in the field of food adulteration detection by virtue of its advantages of non-destructiveness, high sensitivity [115], and specific response to low-frequency vibration of molecules. The problem of food adulteration not only jeopardizes consumers’ rights and interests but also threatens public health and safety [116], especially in high-end agricultural products (e.g., high-quality rice, functional food ingredients). Traditional detection methods, such as sensory evaluation and chromatographic analysis, have limitations such as low efficiency, high cost, or sample destruction [117], while terahertz spectroscopy provides a new idea for fast and accurate adulteration identification by combining with chemometric algorithms [118].

In 2019, Chao Li et al. [82] systematically evaluated the classification performance of terahertz spectroscopy combined with multimodal algorithms [119] for the first time for the rice adulteration problem. In the study, aromatic Wuzhang rice and ordinary long-grain rice were selected as samples, and the mixed powder was prepared and pressed by ball milling, and the transmission spectra were collected in the range of 0–6.4 THz using a terahertz time-domain spectroscopy system (TERA K15). To eliminate environmental interference, the team preprocessed the spectra using SNV, baseline correction (BC), and 1st derivative, and introduced PCA downscaling to extract key features. Based on this, the classification performance of partial least squares discriminant analysis (PLS-DA), support vector machines (SVM), and backpropagation neural networks (BPNN) was compared [120]. The experiments show that the accuracy of the SVM model preprocessed by 1st derivative is as high as 97.33% in the prediction set of five types of adulteration ratios (0%, 33.3%, 50%, 66.7%, and 100%), which is significantly better than other methods. This study not only verifies the feasibility of terahertz spectroscopy in distinguishing the mixing proportions of high- and low-quality rice but also reveals the key influence of preprocessing techniques and algorithm selection on the model performance, which lays a foundation for the subsequent research on complex adulteration scenarios.

In 2021, Li Bin’s team [121] further extended the terahertz spectroscopy technology to the field of kudzu flour adulteration detection. Aiming at the difficult problem that kudzu flour is often mixed with multi-component starches such as lotus root flour, potato flour and rice flour, the study used a TAS7400TS spectrometer from Edwin, Japan, to collect data in the 0.1–7 THz frequency band and focus on the absorption coefficients of 0.5–3.0 THz to analyze the data. The team first tried full-spectrum modeling and found that the PLS-DA model [121] had a high misclassification rate of 57.6%, whereas the least-squares support vector machine decision analysis (LS-SVM-DA) misclassification rate was reduced to 13.3% using both linear kernel (LIN) and RBF. To further improve the accuracy, the study introduces the successive projection algorithm (SPA) and UVE to screen the feature wavelengths. The results show that the PLS-DA misclassification rate is reduced to 33.3% after SPA optimization, while the LS-SVM-DA model with UVE combined with RBF kernel has a misclassification rate of only 6.7%, which is significantly better than other combinations. This result not only confirms the importance of variable selection for complex adulteration detection but also clarifies the advantage of UVE in eliminating spectral redundant information, which provides methodological support for accurate discrimination of multi-component adulteration.

In a similar study, LI Bin’s team [122] also explored the co-optimization strategy of GA and support vector machine. Aiming at the difficult problem of choosing c and g in SVM [123], the study conducted global optimization search by GA, set the population size to 20, the crossover rate to 0.4, and the mutation rate to 0.01, and identified the optimal parameter combinations (c ∈ [0.01, 100], g ∈ [0.01, 100]) using five-fold cross-validation. The optimized SVM model improves the classification accuracy on the kudzu flour adulteration dataset by about 8% compared with the traditional grid search method, which verifies the efficiency of the intelligent algorithm in parameter tuning. In addition, the team compared the hidden layer node self-generation method of BPNN and found that the initial hidden layer node, set to 1, was dynamically expanded to an optimal structure with 5 nodes, effectively avoiding overfitting, and the prediction set accuracy reached 94.5%. These technical innovations jointly enhance the robustness and generalization performance of terahertz spectral models [124]. Table 6 summarizes the applications of terahertz spectroscopy for adulteration detection.

Despite the significant progress in the above studies, terahertz spectroscopy still faces challenges in food adulteration detection. For example, in the rice study, samples need to be ground and pressed, thereby limiting their applicability for in situ detection. Although the kudzu flour experiment verified the multi-component adulteration model, it did not involve the sensitivity test for lower adulteration percentage (e.g., 5–10%). With the decrease of hardware cost and the acceleration of algorithm innovation, terahertz spectroscopy is expected to become an indispensable analytical tool in the food quality and safety monitoring system [125].

### 3.5. Identification of Varieties

Terahertz spectroscopy, as an emerging non-destructive means of detection, has demonstrated significant application potential in the field of food variety identification. Accurate identification of food varieties is of great significance in ensuring food safety, maintaining market order, and satisfying consumers’ right to know. For example, the protein content of different wheat varieties directly affects their processing use and economic value, the compositional characteristics of different tea varieties directly affect their flavor quality and market positioning [126], and the rapid identification of GM crops is related to biosafety regulation. In the following, the specific applications and technological advances of terahertz spectroscopy in varietal identification will be summarized from the perspectives of different food categories and research methods.

For grain variety identification, Chen et al. [127] earlier explored the possibility of combining THz-TDS with deep learning. They took 12 wheat varieties (strong, medium, and weak gluten) as the study object, obtained the absorption coefficient spectra in the 0.2–1.0 THz band by Fourier transform [128], and screened the feature spectra using CARS. The study compares the performance of four models, SVM, LS-SVM, BPNN, and CNN. The results show that the CNN model achieves 98.7% and 97.8% accuracy in the calibration and prediction sets, respectively, with a false identification rate of only 2.2%, which significantly outperforms traditional machine learning methods. This study validates for the first time the high efficiency of fusion of terahertz spectroscopy and deep learning in wheat variety identification and lays the foundation for the subsequent application of complex models.

The identification of genetically modified (GM) crops remains a core issue in food safety regulation. In 2017, Hu et al. [129] developed a joint model based on sparse representation (SR) and RF for GM rice seeds. Experiments were conducted using 200 samples (half of GM and half of non-GM), and an overcomplete dictionary was constructed by the K-SVD algorithm, and the sparse-coded features were extracted and then input to the RF classifier. The results show that the accuracy of the SR-RF model is 95% and 100% in the prediction and calibration sets, respectively [111], which outperforms the traditional LS-SVM method (with about a 5% decrease in accuracy). This study innovatively introduced sparse representation into terahertz data processing, solved the model instability problem caused by high-dimensional spectral features, and provided a new idea for rapid screening of genetically modified seeds. In the same year, Liu et al. [130] extended the study to oil products and established the continuous projection algorithm-linear discriminant analysis (SPA-LDA) with PLS-DA model by combining SG derivative preprocessing and PCA [77] with GM cottonseed oil. Validated by three sample division methods, namely Kennard-Stone, Duplex, and random, the SPA-LDA model has the highest classification accuracy in the test set, confirming that the feature wavelength selection can effectively improve the identification efficiency [131].

In the field of oil identification, another study by Liu [132] focused on genetically modified corn oil. The study compared the performance of PLS-DA models based on physicochemical data and terahertz spectral data. The experiment involved 50 samples (20 genetically modified and 30 non-genetically modified), revealing that the model based on physicochemical data had limited classification accuracy, whereas the model using terahertz spectral data achieved a prediction accuracy of 98.7%. This result highlights the high sensitivity of terahertz technology to microscopic compositional differences, and even if the physicochemical indexes of GM oils overlap with those of conventional oils, accurate differentiation can still be achieved through spectral features.

The identification of coffee varieties and roasting degree is an emerging direction in recent years. In 2024, Huang et al. [133] utilized terahertz spectroscopy combined with PCA [134] and multiple classification algorithms to classify three coffee varieties (Catimor, Typica 1, and Typica 2) and three degrees of roasting (light, medium, and dark) of the same variety. Experiments show that the linear discriminant (LD) model performs best after PCA dimensionality reduction, achieving 100% accuracy in varietal identification within 20 ms. For roasting degree identification, the model also achieves 100% accuracy for all three roasting levels of Typica 1, with a processing time of 25 ms. The study confirms that the terahertz technology not only distinguishes genetic differences among varieties but also captures chemical changes such as oxidation of oils, coking of sugars, etc., during roasting, and provides a good solution for the classification of coffee varieties. Terahertz technology not only distinguishes genetic differences between varieties but also captures chemical changes during the roasting process, such as oxidization of oils and sugars, and caramelization, providing an efficient tool for coffee quality control.

From the perspective of technological evolution, the application of terahertz spectroscopy in food identification presents two major trends: first, the transition from traditional machine learning to deep learning [135], such as the CNN model that significantly improves the classification accuracy of wheat varieties; and second, the optimization of feature extraction methods, such as SR, weighted discriminant analysis, etc., which gradually solve the problem of high-dimensional data redundancy and category imbalance. In addition, the research scope was extended from basic grains to complex matrices such as oils and coffee, which verified the universality of the technology. Table 7 summarizes the application of terahertz spectroscopy for species identification.

### 3.6. Nutrient Testing

Nutritional composition testing of food is a key link to ensure healthy diets and promote the upgrading of the food industry. Its importance lies not only in meeting consumers’ precise nutritional needs—such as for gluten-free or low-lactose products—but also in ensuring food quality control, calibrating functional ingredients, and addressing public health concerns including chronic disease prevention functional ingredient calibration, and the prevention and control of chronic diseases and other public health issues. With the global growth of diet-related metabolic diseases (e.g., diabetes, lactose intolerance), Terahertz spectroscopy technology not only avoids sample loss and reagent contamination risk of traditional methods by virtue of non-contact and label-free detection, but also provides real-time online quality control means in high-end manufacturing fields such as nutritional fortification of infant formula and precision formulation of specialty foods, and promotes the development of food testing from laboratory lag to digital management of the whole industrial chain. It further facilitates the transition of food testing from delayed laboratory sampling to integrated digital management across the entire industrial chain.

In 2020, Yuying Jiang et al. [136] proposed a quantitative analysis method based on the fusion of terahertz spectroscopy and imaging data to address the problem of maltose content changes due to germination during wheat storage [136]. The research team collected data by preparing mixed samples of wheat starch and polyethylene containing 0–25% maltose and utilizing a reflective THz-TDS and imaging system covering a spectral range of 0.1–3.5 THz. To overcome the limitations of a single data source, they innovatively combined spectral absorption coefficients with spatial features of the images and constructed a Boosting integrated learning framework based on a least squares support vector machine (LS-SVM) model, with dimensionality reduction performed using principal component analysis (PCA). The model parameters (e.g., regularization parameter C and kernel function parameter γ) are automatically optimized by designing an iterative termination index based on structural risk minimization. The experimental results show that the prediction performance of the data fusion model is significantly better than that of the single spectral or image model [137], with the RMSE of the calibration and prediction sets reduced to 0.12 and 0.15, respectively, and the R^2^ reaches more than 0.96. In addition, the prediction error of the model for four wheat samples of unknown concentration was less than 5%, which verified its usefulness in early sprouting detection and provided technical support for reducing post-production losses of grain.

In 2021, focusing on the needs of gluten-allergic populations, the team of Qingxia Li [138] explored the feasibility of terahertz spectroscopy for the detection of trace wheat gluten in potato starch [139]. In the study, a mixed sample containing 1.3–100% gluten was used to acquire the time-domain signal using a transmission THz-TDS system, and the absorption coefficients were extracted by Fourier transform in the 0.2–1.6 THz frequency band. To cope with the scattering interference of the powder sample, the team introduced polyethylene (PE) as a filler to improve the signal stability. At the data processing stage, SNV was used to eliminate baseline drift [140], followed by Gaussian process regression (GPR) and linear SVM models, respectively [138]. The findings indicated that the GPR model accurately predicted low gluten concentrations (1.3–20%), achieving an R^2^ of 0.859 and an RMSE of 0.070, outperforming the SVM model, which recorded an R^2^ of 0.715 and an RMSE of 0.101. Further analysis revealed a strong correlation between absorption peaks near 1.37 THz and gluten concentration, suggesting that this band may correspond to gluten protein-specific molecular vibrational modes of gluten proteins. This study is the first to apply THz technology to gluten detection, providing a rapid screening tool for gluten-free food certification.

In the same year, Xiao Wei et al. [141] successfully conducted a rapid quantitative analysis of soybean protein content using terahertz spectroscopy. In the study, 75 batches of soybean samples (30–50% protein content) were collected, 225 experimental samples were prepared by grinding and pressing, and 0.1–1.5 THz absorption spectra were acquired using a T-SPEC-type terahertz system. Comparison of the eight preprocessing methods revealed that auto-scaling could effectively improve the performance of the PLS model, with the Rp increasing from 0.9118 to 0.9282. The artificial bee colony optimization support vector regression (ABC-SVR) algorithm was further employed to screen the radial basis kernel function parameters (c = 3.2, g = 0.8), and the constructed model reached an Rp of 0.9659 in the prediction set with an RMSEP of 1.3085%, and the relative standard deviation (RSD) was only 3.5334%.

Terahertz spectroscopy has demonstrated high sensitivity and utility in the quantitative detection of nutrients such as carbohydrates and proteins when combined with advanced data processing algorithms (e.g., ensemble boosting, GPR, and characteristic wavelength selection). Table 8 summarizes the applications of terahertz spectroscopy for nutrient testing. However, environmental interferences (e.g., moisture, lipids), overlapping characteristic peaks, and high instrumentation costs remain key obstacles to its large-scale application. Future research needs to further explore multimodal data fusion, deep learning model optimization [142], and the development of portable THz devices to promote the full-scale application of this technology in food quality inspection [20].

## 4. Challenges and Future Developments

Although terahertz spectroscopy has demonstrated significant advantages in food quality testing (e.g., high sensitivity, non-destructiveness, and rapid detection capability), it should be noted that direct comparative data between terahertz analysis and other conventional methods are more limited in the current literature. For example, in the detection of pesticide residues, the detection limit of THz-FDS for organophosphorus pesticides was reported to be as low as nanograms, but no comparison data with methods such as GC-MS were provided. Similarly, in the detection of melamine in milk powder, the sensitivity of THz-TDS reached the ppm level, but a quantitative comparison with conventional methods was lacking. Future studies should further focus on the comparative validation of terahertz technology with mature analytical methods in order to comprehensively assess its reliability, precision, and accuracy and to promote the standardized application of this technology.

In addition, even with the results achieved in experimental studies, terahertz spectroscopy still faces many limitations in its practical application, and its practical use in industrial food inspection workflows is still hindered by significant technical and non-technical challenges. Key barriers include pronounced spectral interference from water, the lack of comprehensive spectral databases for complex food matrices, high instrumentation costs, and insufficient regulatory endorsement. These limitations collectively impede the transition of THz spectroscopy from laboratory-based research to real-world industrial applications, particularly in scenarios that demand continuous, in-line inspection on production lines.

### 4.1. Physical and Technical Limitations in Complex Matrices

One of the most significant technical limitations of terahertz spectroscopy arises from its strong interaction with water molecules [143]. Within the frequency range of 0.5–3 THz, liquid water exhibits substantial absorption, which attenuates the transmission of target signals and obscures critical chemical fingerprint information. This limitation is especially pronounced when analyzing high-moisture food products such as fresh produce, meat, and dairy. For instance, Qi Liang et al. [144] demonstrated in their study on the quantitative prediction of pork freshness using THz spectroscopy that the spectral signal experienced considerable attenuation across the 0.2–2.0 THz range as sample moisture content increased. Although the application of a BP-AdaBoost model enhanced prediction accuracy (R^2^ = 0.84), it remained challenging to fully mitigate the masking effects of moisture on characteristic spectral features.

In addition, the complexity of food matrices can lead to superposition, scattering, and nonlinear interference of terahertz spectral signals, thereby further compromising analytical accuracy. Jördens et al. [145], in a study employing pulsed THz spectroscopy to detect foreign objects in chocolate, observed that localized variations in chocolate thickness caused a double-peak structure in the THz pulse signal—indistinguishable from that produced by actual foreign objects—resulting in a high rate of false positives. While the false alarm rate was notably reduced by incorporating height correction through sensor fusion techniques, detection challenges persisted for samples packaged in aluminum foil as well as for foreign objects with compositions similar to that of chocolate.

To address the aforementioned challenges, researchers have proposed a series of technical strategies to enhance the applicability of terahertz spectroscopy in complex food detection. For high-moisture samples, moisture-induced background interference is often mitigated using empirically calibrated compensation models. These models improve the identifiability of target signals by capturing absorption intensities of samples with varying water content across specific frequency bands and constructing regression models for moisture correction during subsequent analysis. For example, Liang et al. [144] successfully applied this approach to achieve accurate modeling for pork freshness evaluation. In another study on aflatoxin B1 detection in peanut oil, Chen et al. [35] employed a combination of second-order derivative preprocessing and sparse multilinear regression to enhance signal resolution under high-humidity conditions, resulting in an R_C_ of 0.96 and an R_CV_ of 0.93.

To overcome spectral overlapping and scattering caused by complex food matrices, multispectral fusion has emerged as an effective strategy. The combination of THz spectroscopy with mid-infrared (MIR) or Raman spectroscopy leverages the complementary responses of each technique to different molecular vibrational modes, enabling higher-dimensional information acquisition and more accurate component identification. Although such fusion approaches remain limited in the field of food quality assessment, they have demonstrated promising performance and practical feasibility in related areas such as materials science and pharmaceutical analysis.

In summary, water absorption and complex matrix interference remain the primary obstacles limiting the widespread application of terahertz spectroscopy in practical food detection scenarios. The development of robust moisture compensation models, along with the integration of multi-source spectral fusion strategies, can effectively enhance the spectral resolution and anti-interference capability of THz systems for target analytes. These advancements offer more reliable technical support for the rapid and non-destructive detection of various food products, including dairy, meat, fruits, and edible oils.

### 4.2. Barriers to Industrial Implementation

Although terahertz spectroscopy demonstrates considerable potential in laboratory research, its practical implementation in industrial settings is still impeded by numerous non-technical challenges, which significantly restrict its path toward commercialization.

A major obstacle lies in the lack of regulatory recognition. At present, global food safety and quality control frameworks have yet to formally acknowledge THz technology. Regulatory agencies such as the Codex Alimentarius, EFSA, and FDA have not issued any official guidelines or certification standards for the application of THz spectroscopy in food analysis. This regulatory gap prevents its inclusion in statutory inspection procedures [146] and limits its acceptance as valid analytical evidence in international food trade, thereby undermining its industrial feasibility [147].

Another critical issue is the absence of a standardized technical system. Compared with established techniques such as HPLC and NIR, which have been widely adopted in the food industry, THz systems still lack unified protocols in areas such as instrument calibration, data processing, sample preparation, and presentation [148]. Significant discrepancies in system architecture, spectral resolution, and algorithm design persist, leading to poor reproducibility across different laboratories. Furthermore, the scarcity of standardized reference materials and comprehensive public spectral databases for food testing hinders the translation of research findings into practical tools and limits the establishment of cross-institutional data comparability.

In addition to these institutional constraints, cost, reproducibility, and portability are key extra-technical factors that currently restrict the real-world deployment of THz spectroscopy [148]. Among these, cost is particularly prohibitive. The core components of THz systems—such as high-frequency sources and ultra-sensitive detectors—require complex manufacturing and precise control, resulting in significantly higher costs than conventional analytical instruments. This creates a high entry barrier, especially for small and medium-sized enterprises. Reproducibility is also a significant concern. THz signals are highly sensitive to environmental variables, such as temperature and humidity, as well as sample heterogeneity [149]. Consequently, the same sample analyzed under different conditions or by different laboratories may yield inconsistent results.

Portability and field-deployability further limit practical application. Most current THz instruments are benchtop systems with large physical footprints and high sensitivity to external conditions, making them unsuitable for on-site use in complex, dynamic industrial environments [150]. For food industry applications, practical deployment depends heavily on mobility, environmental robustness, and rapid response capabilities [151]. However, existing THz systems have not yet achieved sufficient miniaturization, modularity, or environmental adaptability to meet these requirements. Additionally, due to the complexity of system operation and high sensitivity to operator variability, differences in handling can introduce additional uncertainty, further undermining the reproducibility and reliability of detection outcomes.

In summary, despite the unique advantages of THz spectroscopy in non-destructive food analysis, widespread industrial adoption remains hindered by challenges related to regulatory endorsement, technical standardization, cost, and real-world adaptability. Overcoming these barriers is essential to enable the successful transition of THz spectroscopy from laboratory research to routine industrial practice.

### 4.3. Future Development Strategy

To facilitate the transition of terahertz spectroscopy from laboratory research to a practical tool for routine food quality control, it is imperative to establish a systematic and time-bound development roadmap. Given the current state of technological immaturity, lack of regulatory alignment, and fragmented practical applications, we propose a four-phase strategic framework spanning 2025 to 2030. This framework is designed to guide the gradual progression of THz technology toward standardization, engineering implementation, and industrial adoption.

Phase I: System Optimization (2025–2026)

The initial phase focuses on overcoming limitations associated with core hardware components. Miniaturization and cost reduction of terahertz modules are essential, particularly through the development of compact solid-state sources such as optically hybridized systems and quantum cascade lasers (QCLs), which offer high spectral purity and tunability across the 0.1–3 THz frequency range [152]. Enhancing detector sensitivity is also critical and can be achieved using advanced materials such as cryogen-free wavelength meters or graphene-based sensors [153], which enable higher signal-to-noise ratios (SNR) without requiring cryogenic cooling. Furthermore, commonly overlooked issues such as thermal drift and mechanical vibrations should be addressed through the implementation of robust system stabilization techniques to ensure measurement repeatability under real-world conditions.

Phase II: Validation and Benchmarking (2026–2027)

Once the system has been stabilized, establishing analytical credibility becomes essential. Cross-laboratory intercomparison studies involving representative food matrices (e.g., oils, grains, dairy powders) should be conducted under harmonized protocols. These studies are critical for defining baseline performance metrics such as limits of detection (LOD), reproducibility, and matrix effect compensation strategies. Ideally, these metrics should align with international standards such as the Codex Alimentarius or ISO 21570 [154]. Concurrently, a globally accessible open spectral database should be developed, comprising annotated terahertz spectra of contaminants, adulterants, and nutrients across a wide range of food categories. To ensure data interoperability, accompanying metadata should include detailed information on sample preparation procedures, environmental conditions, and spectrometer specifications.

Phase III: Regulatory Engagement (2027–2028)

With established performance indicators, targeted regulatory integration becomes imperative. Comprehensive validation dossiers—including system specifications, statistical analyses, and field trial data—should be submitted to regulatory authorities such as EFSA, FDA, and other relevant national agencies. Conducting pilot trials of terahertz protocols within controlled regulatory environments (e.g., import quarantine stations, grain elevators) can effectively demonstrate their practical applicability.

Phase IV: Industrial Integration (2028–2030)

In this final phase, terahertz spectroscopy will be integrated into industrial food processing workflows. THz instruments will be adapted for in-line or on-line deployment, enabling continuous monitoring [69,155] during critical operations such as sorting, moisture regulation, packaging, and storage. To enhance reliability, THz sensors can be combined with complementary techniques such as near-infrared (NIR) or visual imaging to provide comprehensive assessments of chemical and physical properties [156]. This multimodal approach facilitates the detection of issues including adulteration, microbial spoilage, and inconsistent mixing [157].

Spectral data will be processed by embedded software or transmitted to centralized systems for automated quality control, allowing for early warning without interrupting production [158,159]. Correlating THz measurements with batch records will improve traceability and support food safety auditing. This phase marks the completion of the transition from laboratory-based testing to widespread industrial application, positioning THz spectroscopy as a critical technology for smart food manufacturing.

Figure 8 illustrates the strategic framework, with each phase building upon the achievements of the previous one. This roadmap offers a practical, time-bound pathway to overcoming technical, regulatory, and operational challenges, ultimately facilitating the broad adoption of terahertz spectroscopy within the food industry.

In conclusion, the advancement of terahertz spectroscopy technology depends not only on innovations in hardware and algorithm optimization but also on the establishment of a closed-loop ecosystem that integrates technology, applications, and standards oriented toward practical needs. Realizing such a system necessitates alignment with international regulatory frameworks and method validation protocols. Standards such as ISO 22000:2018 for food safety management systems, the Codex Alimentarius guidelines for national food control, and the European Food Safety Authority’s criteria for analytical method validation provide foundational references for this process. Concurrently, transnational initiatives, including METROFOOD-RI and the Food Integrity project, offer platforms for harmonizing THz applications across diverse food matrices and regulatory jurisdictions [160]. With the continued advancement of AI-driven approaches and interdisciplinary integration, terahertz spectroscopy is poised to become a core tool for food quality and safety monitoring, ultimately driving the intelligent and sustainable transformation of the global food industry.

## Figures and Tables

**Figure 1 foods-14-02199-f001:**
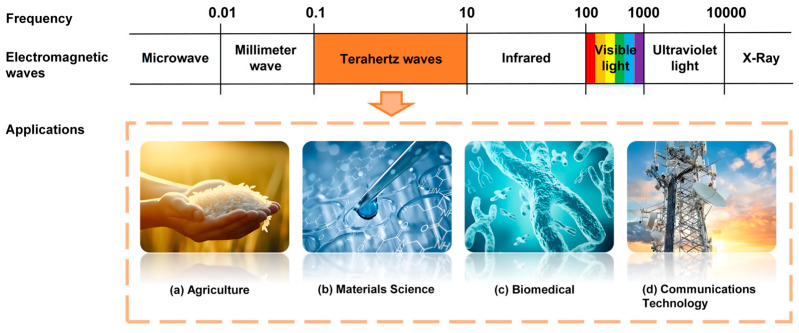
Location of the terahertz band in the electromagnetic spectrum and application areas of terahertz spectroscopy technology. (**a**) Agricultural products quality testing field; (**b**) Materials science field; (**c**) Biomedical field; (**d**) Communications technology field.

**Figure 2 foods-14-02199-f002:**
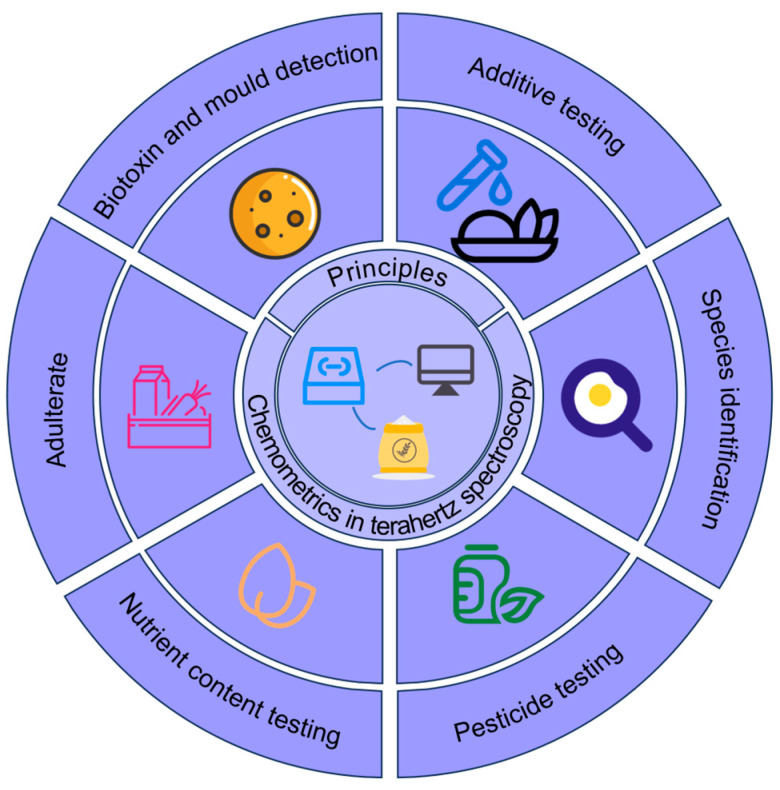
Application of terahertz spectroscopy for food quality testing.

**Figure 3 foods-14-02199-f003:**
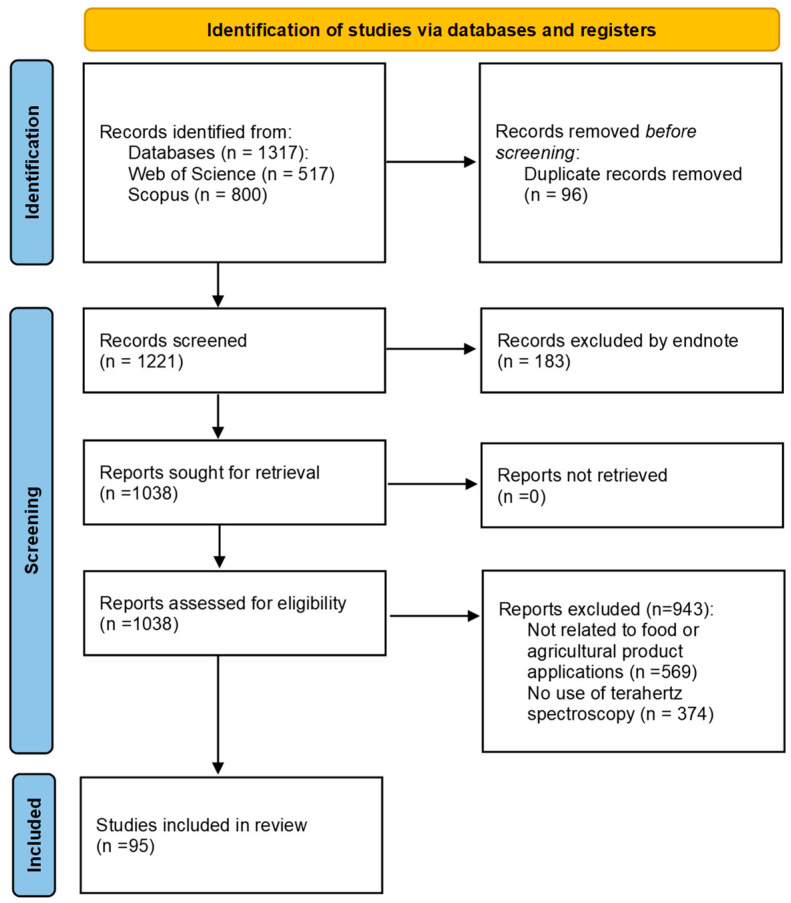
PRISMA literature screening flowchart.

**Figure 4 foods-14-02199-f004:**
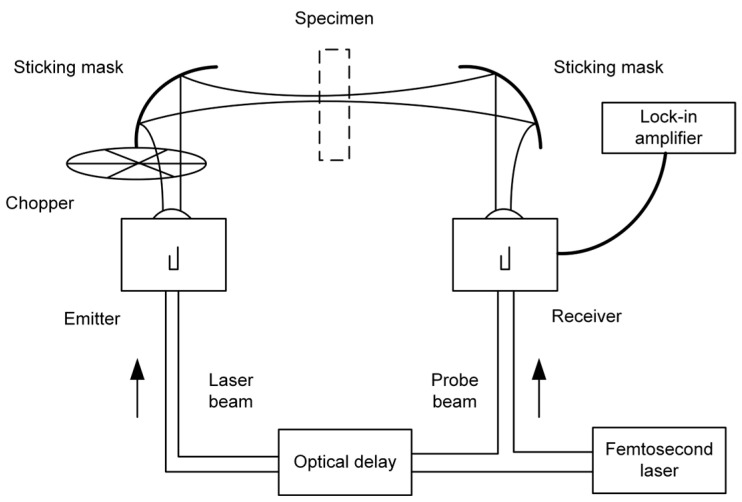
Transmissive terahertz system. The arrows indicate the directions of the laser beam, probe beam, and signal transmission.

**Figure 5 foods-14-02199-f005:**
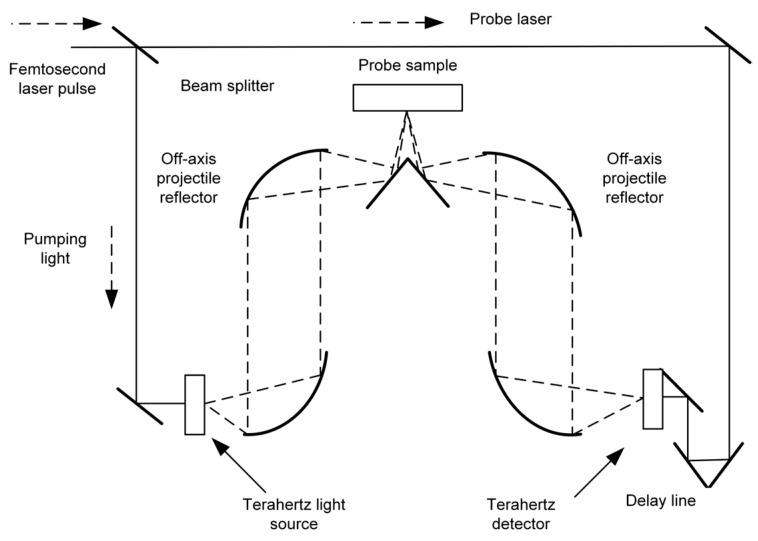
Reflective terahertz system. Dashed arrows indicate the paths of optical signals, including the femtosecond laser pulse, pumping light, and probe beam. Solid arrows represent the propagation and reflection of terahertz radiation from the source, via the sample surface, to the detector.

**Figure 6 foods-14-02199-f006:**
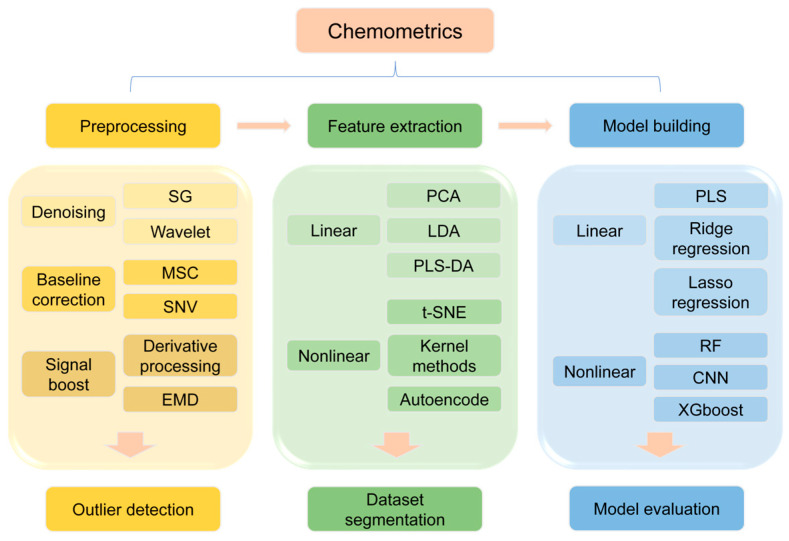
Flow and algorithms for chemometric processing of spectral data.

**Figure 7 foods-14-02199-f007:**
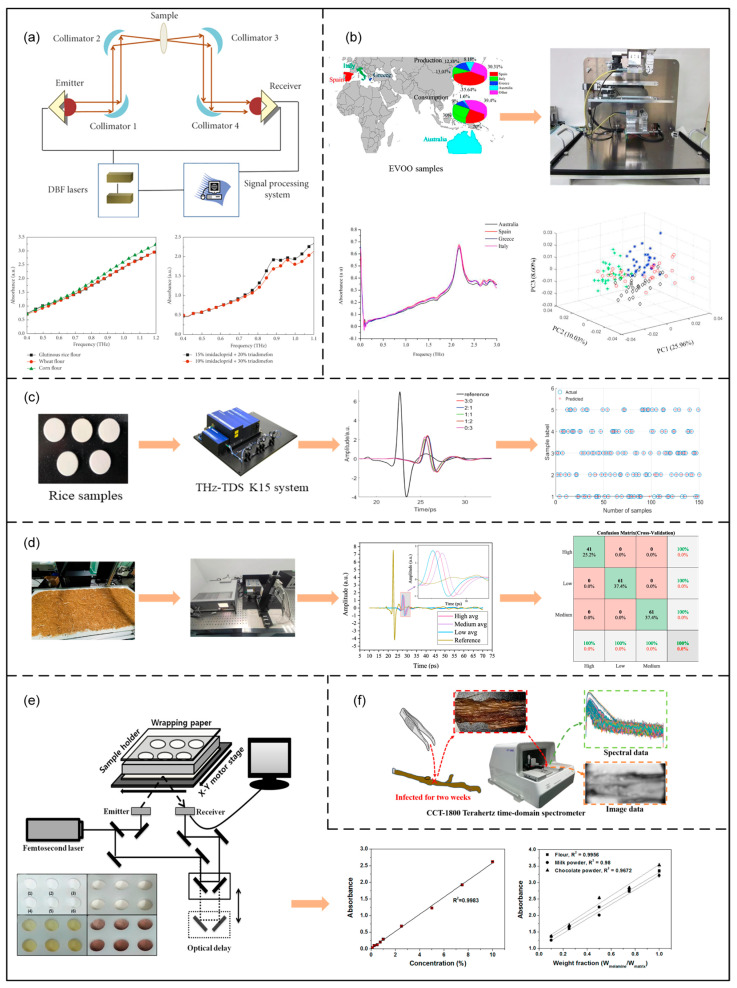
(**a**) Terahertz spectroscopy for pesticide detection [79]; (**b**) Terahertz spectroscopy to identify the origin of olive oil. EVOO samples from Spain (○), Italy (+), Greece (*), Australia (◊). PCA plot shows absorbance-based distribution [80,81]; (**c**) Terahertz spectroscopy to identify rice adulteration [82]; (**d**) Terahertz time-domain spectroscopy to identify the quality of wheat gluten [83]; (**e**) Terahertz time-domain spectroscopy for the detection of melamine in food products [84]; (**f**) Terahertz spectroscopy for the detection of apple rot disease [85].

**Figure 8 foods-14-02199-f008:**
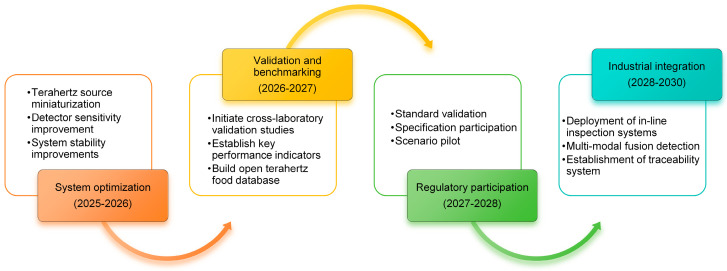
Strategic map of future directions of terahertz spectroscopy in food quality testing.

**Table 1 foods-14-02199-t001:** Characteristics of traditional inspection techniques.

Detection Technology	Technical Characteristics	Detection Time	Advantage	Disadvantage
HPLC	Highly efficient separation	Minutes to tens of minutes	High sensitivity for multi-component analysis	High cost and complex pre-treatment
GC	Volatility analysis	Minutes to 20 min	High sensitivity to volatile components, clear structural information	Requires derivatization and cumbersome sample pretreatment
ELISA	Bioimmune recognition	1–3 h	Highly specific, rapid screening for low concentration indicators	High antibody quality dependence and risk of cross-reactivity
PCR	Molecular detection	2–4 h	High specificity for microbial/transgenic testing	Complex process, strict control of contamination, specialized equipment required
TA	Stoichiometry	Manual: 5~20 min Automatic: 3~10 min	Simple instrument, low cost, suitable for acid-base, salinity and other routine testing	High human error and lower accuracy than instrumental methods
UV–Vis	Absorbance measurement	5–30 min	Fast, low equipment cost, easy to use for color and characterization quantification	Sensitive to sample matrix interference, not suitable for complex systems

HPLC, High-performance liquid chromatography; GC, Gas chromatography; ELISA, Enzyme-linked immunosorbent assay; PCR, Polymerase chain reaction; TA, Titration analysis; UV–Vis, Ultraviolet–visible spectroscopy.

**Table 2 foods-14-02199-t002:** Characteristics of emerging non-destructive testing technologies.

Detection Technology	Wavelength/ Frequency	Brief Outline of Principle	Detection Limit	Advantage	Disadvantage
NIR	780–2500 nm	Molecular vibration absorption	%	Fast, non-destructive, simultaneous measurement of components	Sensitive to complex matrices, model-dependent
THz	3 mm–30 µm	Low-frequency resonance absorption	ppm	Wearable packaging, identifies adulteration, non-ionizing and harmless	Heavy water interference, limited sensitivity and penetration
Raman	532–1064 nm (excitation light)	Molecular vibration induced scattered radio frequency shifts	ppm	Highly sensitive, resistant to water interference, rapid detection	Fluorescence interference, high equipment cost
X-ray	0.01–10 nm	Penetration and absorption imaging	µg–mg	Industrially proven, efficient identification of hard foreign objects	Weak recognition of low-density foreign objects such as mixed plastics
CV	Visible light 400–700 nm	Visual feature extraction and recognition	µm	Real-time online sorting at low cost	Vulnerable to light and background
UT	16 kHz–20 MHz	Echo detection of sound waves	mm	Detects sealing defects, internal tissue structure	Requires coupling agent, fast signal attenuation, poor results with complex samples
MW	30–300 GHz	Dielectric response to EM waves	%	Secure, low cost, penetrating packaging, suitable for online	Low resolution, difficult to identify small particles or trace adulteration
NMR	43–100 MHz	Nuclear spin resonance detection	%	Non-destructive quantification, suitable for water/lipid analysis	Expensive equipment, insufficient sensitivity for trace adulteration detection
HSI	400–2500 nm	Combined spectral and spatial analysis	ppm	Graphically rich, detects spoilage, adulteration, spotting	Large data volumes, complex models, expensive equipment

NIR, Near-infrared spectroscopy; THz, Terahertz spectroscopy; Raman, Raman spectroscopy; X-ray, X-ray inspection; CV, Computer vision; UT, Ultrasonic testing; MW, Microwave sensing; NMR, Nuclear magnetic resonance spectroscopy; HIS, Hyperspectral imaging.

**Table 3 foods-14-02199-t003:** Applications of terahertz spectroscopy for pesticide detection.

Food	Target of Detection	Typology	Arithmetic	Outcome	Reference
Polyethylene and glutinous rice flour	Imidacloprid	Pesticide detection	PLS	Relative error of prediction < 5.00% RMSEP = 0.70%	[86]
Flour	Imidacloprid Carbendazim	Pesticide detection	MSBC-PLS-Voigt	R_P_ = 0.9914 R_C_ = 0.9957	[89]
Wheat flour	6-BA 2,6-D Imidacloprid	Pesticide detection	BPNN	R_P_ = 0.9913 R_P_ = 0.9948 R_P_ = 0.9923	[93]

**Table 4 foods-14-02199-t004:** Application of terahertz spectroscopy for additive detection.

Food	Target of Detection	Typology	Arithmetic	Outcome	Reference
Wheat flour	Benzoic acid	Additive detection	CARS-PCA-LS-SVM	R_P_ = 0.9956 RMSEP = 0.64%	[97]
Coumarin-based food additives	Coumarin-based food additives	Additive detection	P-t-SNE-DEGWO-SVM	R_P_ = 0.9861	[98]

**Table 5 foods-14-02199-t005:** Application of terahertz spectroscopy for biotoxin and mold detection.

**Food**	**Target of Detection**	**Typology**	**Arithmetic**	**Outcome**	**Reference**
Acetonitrile solution	AFB1	Toxins and mildew	PLSR (High concentration) RBF-SVM (low concentration)	R = 0.9780	[107]
				R = 0.9376 RMSE = 0.0779	
Peanut oil	AFB1	Toxins and mildew	Second derivative-SMLR	R_CV_ = 0.9309 R_C_ = 0.9639	[35]
Black tea	OTA	Toxins and mildew	/	RMSEC = 0.2800 RMSEP = 0.3500	[112]

**Table 6 foods-14-02199-t006:** Application of terahertz spectroscopy for adulteration detection.

Food	Target of Detection	Typology	Arithmetic	Outcome	Reference
Rice	Different mixed proportions of rice	Adulteration	1st derivative-PCA-SVM	Accuracy = 97.33%	[82]
Kudzu powder	Lotus root powder and potato powder	Adulteration	UVE-LS-SVM-RBF-DA	Misjudgment rate = 6.70%	[121]
Panax notoginseng powder	Zedoary turmeric powder Wheat flour	Adulteration	LS-SVM	R_P_ = 0.9015 RMSEP = 0.0723 R_P_ = 0.9305 RMSEP = 0.0677	[122]
	Rice flour		PLS	R_P_ = 0.9424 RMSEP = 0.0601	

**Table 7 foods-14-02199-t007:** Application of terahertz spectroscopy for species identification.

**Food**	**Target of Detection**	**Typology**	**Arithmetic**	**Outcome**	**Reference**
Wheat grain	Strong-gluten Medium-gluten Weak-gluten	Variety identification	SNV-CNN	Misjudgment rate = 2.20%	[127]
Rice seeds	Gene	Variety identification	SR-RF	Accuracy = 95.00%	[129]
Cottonseed oil	Gene	Variety identification	PLS-DA	Accuracy = 97.00%	[130]
Corn oil	Gene	Variety identification	PLS-DA	Accuracy = 98.70%	[132]
Coffee beans	Degree of baking	Variety identification	PCA-LD	Accuracy = 100.00%	[133]

**Table 8 foods-14-02199-t008:** Application of terahertz spectroscopy for nutrient detection.

Food	Target of Detection	Typology	Arithmetic	Outcome	Reference
Wheat	Maltose	Nutrient content	PCA-Boosting-LS-SVM	R^2^ > 0.9600 RMSEC = 0.1200 RMSEP = 0.1500	[136]
Potato starch	Wheat gluten	Nutrient content	GPR	R^2^ = 0.8590 RMSE = 0.0700	[138]
Soybeans	Protein	Nutrient content	SNV-second derivative- ABC-SVR	R_P_ = 0.9659 RMSEP = 1.31% RSD = 3.53%	[141]

## Data Availability

No new data were created or analyzed in this study. Data sharing is not applicable.

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
