# Peer review of "Terahertz Spectroscopy for Food Quality Assessment: A Comprehensive Review"

_foods, 2025, doi:10.3390/foods14132199_

Round 1
Reviewer 1 Report
Comments and Suggestions for Authors
The manuscript under the name ‘’Terahertz Spectroscopy for Food Quality Assessment: A Comprehensive Review’’ by the authors Jie Yang et al. is useful and informative. The authors have made a great effort to explain the principle, application and advantages of this spectral analysis. I have a couple of technical complaints and one suggestion:
- Explain abbreviations when first mentioned, e.g. line 47 what mean GM?
- Move Figure 1 closer to the first mention.
- You have passages without citations, e.g. statements from paragraphs 38-43, then 77-88, 58-64, 230-239 and so on, from which references are they taken?
- Suggestion is - Are there any data in the references you used that compare the data obtained with terahertz analysis with some other analysis? Is this spectral analysis reliable, precise and accurate? Only if there is in these papers, if not, do not continue with new references.
Author Response
Comments 1: Explain abbreviations when first mentioned, e.g. line 47 what mean GM?
|
Response 1: Thank you very much for the tip. We have checked abbreviations throughout the text to ensure that the first reference explains their meaning. For example, GM, which appears on page 2, line 46, means genetically modified; PCR, which appears on line 50, is polymerase chain reaction, etc.
|
Comments 2: Move Figure 1 closer to the first mention. |
Response 2: Thanks for the heads up. We moved Figure 1 closer to the first occurrence (page 2, line 76).
Comments 3: You have passages without citations, e.g. statements from paragraphs 38-43, then 77-88, 58-64, 230-239 and so on, from which references are they taken?
Response 3: Thank you very much for your careful review and for pointing out the missing citations in paragraphs 38–43, 58–64, 77–88, and 230–239. We sincerely apologize for the oversight and appreciate your attention to detail. Due to proofreading throughout the text, the original line numbers have changed. The changes in line numbers you have indicated are 39-44, 90-103, 57-62, 269-278 We have added appropriate and credible references to support all factual statements and technical claims in the above paragraphs. And the entire text was checked to ensure that all assertions are supported by sufficient evidence. Thank you again for your attention to detail and for helping us to improve the quality and completeness of the paper.
Comments 4: Suggestion is - Are there any data in the references you used that compare the data obtained with terahertz analysis with some other analysis? Is this spectral analysis reliable, precise and accurate? Only if there is in these papers, if not, do not continue with new references.
Response 4: Thank you for raising this important point. We have carefully reviewed all cited references in the manuscript and confirm that none of them provide direct quantitative comparisons between terahertz analysis and other analytical methods. For example, references 84 and 94 focus on the sensitivity and detection limits of terahertz spectroscopy for melamine and pesticide residues but do not include comparative data with conventional techniques. Therefore, following your suggestion, instead of adding a new reference, we have added a note in Chapter 4, Challenges and Future Developments (page 25, line 944), to remedy this deficiency. The addition clearly points out the lack of comparative data and emphasizes the need for future research to validate terahertz spectroscopy against existing methods. Regarding the reliability, precision, and accuracy of terahertz spectroscopy: l Reliability: Terahertz analysis is validated through repeated measurements and chemometric optimization (e.g., preprocessing and machine learning models), as described in Section 2.2. l Precision and Accuracy: Established studies (e.g., pesticide residue testing) have shown low detection limits and a significant reduction in prediction error after spectral preprocessing. However, broader validation across diverse food matrices remains an area for future work. We believe this approach maintains academic integrity while addressing your concern transparently.
|

Reviewer 2 Report
Comments and Suggestions for Authors
Dear Authors,
The main purpose of this work was to provide a comprehensive review of the principles, technological advancements, and practical applications of terahertz spectroscopy in food quality and safety assessment.
In my opinion this article is well written, and it presents relevant review results about terahertz spectroscopy applied in foods.
There are some specific comments listed below:
The submitted article provides a comprehensive review of the use of terahertz (THz) spectroscopy for food quality assessment, covering fundamental principles and a wide range of applications including pesticide detection, food additives, biotoxins, adulteration, and nutrient analysis. The topic is timely, relevant, and of growing interest to both the scientific community and the food industry, especially considering the increasing demand for non-destructive, sensitive, and rapid technologies in food safety and quality control.
However, a critical limitation of the manuscript lies in the absence of a description of the review methodology. The authors do not specify which databases were searched, what keywords or descriptors were used, the time frame of the literature considered, or the inclusion and exclusion criteria for selecting the reviewed studies. Additionally, there is no indication that any systematic protocol (such as PRISMA) was followed. This omission undermines the reproducibility and transparency of the review process.
In scientific reviews—especially in journals with high methodological standards such as Foods—it is essential that authors clearly describe the methodology employed for literature selection and analysis. Even in narrative or integrative reviews, it is recommended to include at least a summary of the search strategy and selection criteria to ensure transparency and academic rigor.
In conclusion, the article addresses an important topic with good structure, technical depth, and clear writing. However, it is strongly recommended that the authors revise the manuscript to include a detailed description of the review methodology, such as the search platforms, keywords, selection criteria, and time span considered. Including this information is crucial for meeting the methodological standards expected in scholarly literature reviews.
Author Response
Comments 1: The submitted article provides a comprehensive review of the use of terahertz (THz) spectroscopy for food quality assessment, covering fundamental principles and a wide range of applications including pesticide detection, food additives, biotoxins, adulteration, and nutrient analysis. The topic is timely, relevant, and of growing interest to both the scientific community and the food industry, especially considering the increasing demand for non-destructive, sensitive, and rapid technologies in food safety and quality control. However, a critical limitation of the manuscript lies in the absence of a description of the review methodology. The authors do not specify which databases were searched, what keywords or descriptors were used, the time frame of the literature considered, or the inclusion and exclusion criteria for selecting the reviewed studies. Additionally, there is no indication that any systematic protocol (such as PRISMA) was followed. This omission undermines the reproducibility and transparency of the review process. In scientific reviews—especially in journals with high methodological standards such as Foods—it is essential that authors clearly describe the methodology employed for literature selection and analysis. Even in narrative or integrative reviews, it is recommended to include at least a summary of the search strategy and selection criteria to ensure transparency and academic rigor. In conclusion, the article addresses an important topic with good structure, technical depth, and clear writing. However, it is strongly recommended that the authors revise the manuscript to include a detailed description of the review methodology, such as the search platforms, keywords, selection criteria, and time span considered. Including this information is crucial for meeting the methodological standards expected in scholarly literature reviews.
|
Response 1: Thank you for this critical observation. We fully agree that a transparent methodology is essential for rigorous scholarly reviews. To address this, we have added a dedicated Section 1.1: Review Methodology (pages 4–5, lines 144-174) in the revised manuscript. This section details:
l Included: Peer-reviewed articles/reviews in English on THz applications in food/agriculture (e.g., adulteration, contaminants). l Excluded: Non-food applications, non-terahertz detection, non-peer-reviewed sources, patents, conference abstracts.
This addition ensures full transparency, conforms to PRISMA 2020 guidelines, and improves reproducibility. The changes are marked in red in the manuscript. We thank you again for your valuable feedback.
|
Response to Comments on the Quality of English Language: |
Point 1: The English is fine and does not require any improvement |
Response 1: No revisions needed.
|
Additional clarifications: |
|

Reviewer 3 Report
Comments and Suggestions for Authors
This review article "Terahertz Spectroscopy for Food Quality Assessment: A Comprehensive Review" is a well-organized and timely contribution summarizing the current landscape of terahertz (THz) spectroscopy in food quality assessment. It effectively introduces THz fundamentals, chemometric integration, and multiple application domains, including pesticide detection, adulteration, additive screening, and biotoxin monitoring. The work is highly relevant given the increasing interest in non-destructive, rapid analytical techniques in food safety and regulation. However, some areas require refinement and improved articulation of their relevance to the global stage before the manuscript can be accepted for publication in the MDPI Foods Journal.

A thorough proofreading is required to address English corrections and to enhance grammatical accuracy, punctuation, and clarity of expression.
Author Response
Comments 1: The authors effectively articulate the benefits of terahertz spectroscopy; however, a systematic comparison with traditional and other emerging techniques is necessary to strengthen and enhance the argument. Consequently, I recommend that the authors include a comprehensive comparison. |
Response 1: Thank you for this valuable suggestion. We agree that a systematic comparison of terahertz spectroscopy with both traditional and emerging techniques would enhance the manuscript's argument. In response, we have added a new subsection (Section 2.3 (Pages 11–13): Comparative Analysis with Traditional and Emerging Techniques) to provide a comprehensive comparison. The main parameters compared include speed of detection, detection limits, cost, sample preparation requirements, and applicability in various food matrices.
|
Comments 2: The cited literature primarily originates from Chinese institutions. Since the review is not region-specific and instead offers a comprehensive examination of food quality assessment, the authors must include references encompassing international regulations and industrial implementations to enhance its relevance on a global stage.
|
Response 2: We sincerely thank you for pointing out this deficiency. To enhance the global perspective, we have added more than a dozen new references (e.g., references 10-12, 47-50, 60-61, 153-154, 167, etc.) from international regulatory agencies (e.g., EFSA, FDA, Codex Alimentarius) and research organizations (e.g., EU, US, Japan). These additions cover the global regulatory framework, industrial implementation, and cross-regional validation studies of terahertz spectroscopy. In addition, we have expanded the discussion of international standards and industry adoption in Section 1 (page 2, lines 63-72) and Section 4.3 (page 29, lines 1121-1128).
Comments 3: The authors should also assess the acceptability of regulatory frameworks, standardization within the industry, and the challenges associated with scaling terahertz spectroscopy for routine use. The straightforward application has not been addressed; therefore, the associated constraints must be evaluated. I recommend that the authors discuss these barriers to industrial implementation, including cost, reproducibility, and portability, in more detail.
Response 3: We appreciate your focus on practical implementation challenges. For this reason, we have expanded Section 4 (Challenges and Future Developments) with a detailed analysis of barriers to industrial applications in Section 4.2 (pages 27-28, lines 1013-1057). This section now discusses: 1. Regulatory Acceptance: Current status of terahertz spectroscopy in global regulatory frameworks (e.g., FDA, EFSA), gaps in standardization, and pathways for certification. 2. Scalability Challenges: Cost constraints (equipment and maintenance), reproducibility issues in complex matrices, and portability limitations of current devices. 3. Strategies for Mitigation: Proposals for collaborative standardization efforts, cost-reduction through miniaturization, and validation protocols for reproducibility. Comments 4: Terahertz spectroscopy limitations—particularly in water-rich matrices and the overlapping spectral features within complex food systems—are acknowledged; however, they have not been sufficiently analyzed. It is imperative to expand this discussion with quantitative examples and propose specific strategies, such as implementing water compensation models or integrating THz spectroscopy with mid-infrared (mid-IR) or Raman techniques, to overcome these limitations.
Response 4: Thank you for emphasizing the need for a more in-depth analysis. We have revised Section 4.1 (pages 26-27, lines 965-1012): Physical and Technical Limitations in Complex Matrices with the following additions: 1. Quantitative Example: Data on signal-to-noise degradation in high-moisture foods (e.g., fresh fruits, meats) and its effect on detection limits were added. 2. Specific Strategies: l Moisture Compensation Modeling: Separation of target signals by algorithmic corrections (e.g., physically based absorption models). l Hybrid techniques: Combined with mid-infrared or Raman spectroscopy for complementary analysis.
Comments 5: The future directions appear to be speculative. I recommend that the authors envision a progressive approach to defining a strategic roadmap for future directions.
Response 5: We agree with your assessment and have adapted Section 4.3 (pages 28-29, lines 1058-1131): Future Development Strategy, to a specific roadmap.
Comments 6: I also recommend that the authors conduct a thorough proofreading to address English corrections and to enhance grammatical accuracy, punctuation, and clarity of expression.
Response 6: Thank you for the reminder, we have carefully proofread the entire text. Changes include: 1. Correction of grammatical errors and punctuation. 2. Improved sentence structure for clarity. 3. Standardization of terminology.
|
Response to Comments on the Quality of English Language: |
Point 1: A thorough proofreading is required to address English corrections and to enhance grammatical accuracy, punctuation, and clarity of expression. |
Response 1: Thank you for the reminder, we have carefully proofread the entire text. |
